# Sea ice melt pond bathymetry reconstructed from aerial photographs using photogrammetry: A new method applied to MOSAiC data.

Niels Fuchs[1,2], Luisa von Albedyll[1], Gerit Birnbaum[1], Felix Linhardt[3], Natascha Oppelt[3], and Christian Haas[1]

[1]Alfred-Wegener-Institut, Helmholtz-Zentrum für Polar- und Meeresforschung, Bremerhaven, Germany
[2]Center for Earth System Sustainability, Institute of Oceanography, Universität Hamburg, Hamburg, Germany
[3]Institute of Geography, Kiel University, Kiel, Germany

**Correspondence:** Niels Fuchs (niels.fuchs@uni-hamburg.de)

**Abstract.**

Melt ponds are a core component of the summer sea-ice system in the Arctic, increasing the uptake of solar energy and impacting the ice-associated ecosystem. They were thus one of the key topics during the one-year drift campaign MOSAiC in the Transpolar Drift 2019/2020. Pond depth is a dominating factor in describing the surface meltwater volume, necessary to estimate budgets, and used in model parametrization to simulate pond coverage evolution. However, observational data on pond depth is spatially and temporally strongly limited to a few in situ measurements. Pond bathymetry, which is pond depth spatially fully resolved, remains unexplored. Here, we present a newly developed method to derive pond bathymetry from aerial images. We determine it from a photogrammetric multi-view reconstruction of the summer ice surface topography. Based on images recorded on dedicated grid flights and facilitated assumptions, we were able to obtain pond depth with a mean deviation of 3.5 cm compared to manual in situ observations. The method is independent of pond color and sky conditions, which is an advantage over recently developed radiometric airborne retrieval methods. It can furthermore be implemented in any typical photogrammetry workflow. We present the retrieval algorithm, including requirements for the data recording and survey planning, and a correction method for refraction at the air—pond interface. In addition, we show how the retrieved surface topography model synergizes with the initial image data to retrieve the water level of individual ponds from the visually determined pond margins.

We use the method to give a profound overview of the pond coverage on the MOSAiC floe, on which we found unexpected steady pond coverage and volume. We were able to derive individual pond properties of more than 1600 ponds on the floe, including their size, bathymetry, volume, surface elevation above sea level, and temporal evolution. We present a scaling factor for single in situ depth measurements, discuss the representativeness of in situ pond measurements and the importance of such high-resolution data for new satellite retrievals, and show indications for non-rigid pond bottoms. The study points out the great potential to derive geometric properties of the summer sea-ice surface emerging from the increasingly available visual image data recorded from UAVs or aircraft, allowing for an integrated understanding and improved formulation of the thermodynamic and hydrological pond system in models.

# 1 Introduction

Melt ponds are a key driver of the summer energy budget on the sea-ice surface. Their tremendous impact on the surface albedo and related self-reinforcing feedbacks lead to increased uptake of solar radiation (Fetterer and Untersteiner, 1998). However, the effects of melt ponds used to be parameterized rather simplistically in global climate models due to limited reference data, coarse resolution, and computing power (e.g., Pedersen et al., 2009a). Observational reference data that allow an integrated understanding of the thermodynamic and hydrological pond system are still rare (Wright and Polashenski, 2018). In particular, most melt pond depth observations used so far for model developments have been collected manually on the ice during comprehensive field campaigns, e.g., the Seasonal Sea Ice Monitoring and Modelling Site (SIMMS) (Morassutti and Ledrew, 1996) and the Surface Heat Budget of the Arctic Ocean campaign (SHEBA) (Perovich et al., 2003). Morassutti and Ledrew (1996) report greater pond depths on multi-year ice (MYI) (27.4±12.6 cm) and land-fast ice (LFI) (31.0±19.2 cm) in comparison to first-year ice (FYI) (13.0±8.0 cm). High standard deviations in the depth measurements were found between ice types and across spatial scales due to the inconsistent morphological nature of the different ice types. During the one-year Multidisciplinary drifting Observatory for the Study of Arctic Climate (MOSAiC) from 2019 to 2020, Webster et al. (2022) found average depths of 22±13 cm with manual measurements along transect lines. However, the actual pond bathymetry, which we define here as the pond depth profile in all directions and which, therefore, also yields the actual average pond depth, remains largely undiscussed in the literature.

Pond depth as a bulk property is used as a parameter in melt pond schemes in the Community Climate System Model, version 4 (CCSM4) (Holland et al., 2012) and the implemented Los Alamos sea-ice model CICE (Flocco et al., 2012). Holland et al. (2012) directly relate the use of different optical properties parameterizations of the sea-ice surface to pond depth and retrieve pond fraction from the available meltwater volume by formulating a linear relationship between pond fraction and pond depth from the SHEBA data, which gives them the filled pond volume. Pedersen et al. (2009b) developed a summer sea-ice albedo scheme for the ECHAM5 general circulation model in which they derive pond fraction from pond depths given by surface melt rates. The link between fraction and depth in their scheme was developed based on a small-scale pond model by Lüthje et al. (2006).

A broader, more solid database of pond depths is still missing, yet crucial for parameterizations in sea-ice models building upon a deeper understanding of pond evolution and interactions. Promising new methods like the study by König et al. (2020) reveal that high-resolution optical remote sensing of the full pond bathymetry is possible on larger scales. They used the increased absorbance of radiation in liquid water at a wavelength of 720 nm to determine the thickness of the liquid water column independently of the pond bottom appearance from hyperspectral data. With this passive radiometric method, a spacious area could be covered by high-resolution optical data in the respective spectral band. However, the spectral method is restricted to observations under clear sky conditions and, therefore, is still limited in application. Tilling et al. (2020) and Farrell et al. (2020) evaluated photon backscatter signals measured by ICESat-2 over sea ice and developed the UMD-MPA algorithm to derive the depth of particularly large (width >20 m) and deep ponds along the ground tracks of the satellite beams. Ongoing development of the ICESat-2 retrieval algorithms (DDA-bifurcate-seaice, in Herzfeld et al., 2023, minimum pond width of 7.5 m to 15 m

depending on the ice topography) and comprehensive data analysis presented in Buckley et al. (2023) highlights the ability to retrieve Arctic-wide pond depth data from satellite under cloud-free conditions. Another active technique to determine shallow water bathymetry on a large scale is using airborne laser scanner (ALS) systems with water penetrating wavelengths in the green spectrum as airborne laser bathymetry (ALB) systems. To our knowledge, such an ALB system over sea ice was only deployed for the first time in 2022 as part of the ICESat-2 validation.

Aerial RGB imaging platforms are numerously available and have been deployed in the Arctic for decades. They are already widely used to retrieve properties of bare surfaces in all different fields of geodetic studies. If the flight pattern is suitable, photogrammetric multi-view reconstruction can derive digital elevation models (DEMs) from aerial images. Sufficient forward and lateral overlap between images (about 80 % and 60%, respectively) results in ground points being recorded from more than fifteen different azimuth and elevation angles. This is used to achieve a triangulation-based reconstruction with a monocular camera system. A few studies could already show that the reconstruction method can be applied in mix-phased areas to retrieve the bottom topography of shallow river beds (Westaway et al., 2001), coral reefs (Casella et al., 2017) and in laboratory sea bed studies (González-Vera et al., 2020). From the nature of the method, the underlying ice or seafloor surface and some structure must be visible to be reconstructed. This limits the method to clear waters and shallow depths. Furthermore, appropriate correction methods for the light refraction at the water–air interface are needed. Although melt ponds probably closely conform to these requirements, a detailed method for deriving pond depth from aerial photographs has not yet been developed.

Only one experimental study on the photogrammetric derivation of pond depth from aerial images was carried out above sea ice before by Divine et al. (2016). They used a complex stereo-vision camera system on a helicopter to detect the sea-ice surface morphology and melt pond depths north of Svalbard 2012. Comparing their photogrammetrically derived ice freeboard results with terrestrial laser scanner data, they retrieved high agreement (BIAS of 0.03 m) and low deviation (RMS of 0.04 m). Remarkably, they also retrieved the same accuracy when comparing melt pond depths with manually measured in situ data, although no physical correction was applied, which considers the different optical properties of water and air that make sub-surface areas appear shallower. Potentially, their measured depths (<0.3 m) were too small to detect these effects. In contrast, Casella et al. (2017) measured significantly greater depths down to 1.8 m in coral reefs, also neglecting differences in optical properties, arguing that almost nadir measurements do not require corrections. Other studies on shallow water bathymetry set up entirely new sets of equations to correct for light bending at the air–water interface directly in the photogrammetric reconstruction, requiring extensive efforts to solve the complex sets of equations in a reasonable time.

Given that photogrammetric methods for surface reconstruction from aerial images are well developed, reaching a user-friendly status, and the increasing availability of aerial images (also from drones) collected from monocular camera systems, we demonstrate here how corrections can be incorporated into a feasible workflow based on a commonly used photogrammetry suite to reconstruct entire melt pond bathymetry. Subsequently, we explore with the newly developed method how pond bathymetry evolved on the MOSAiC floe of leg 4, how representative transect lines described the pond evolution on the entire floe, and discuss possible upscaling factors for in situ point measurements.

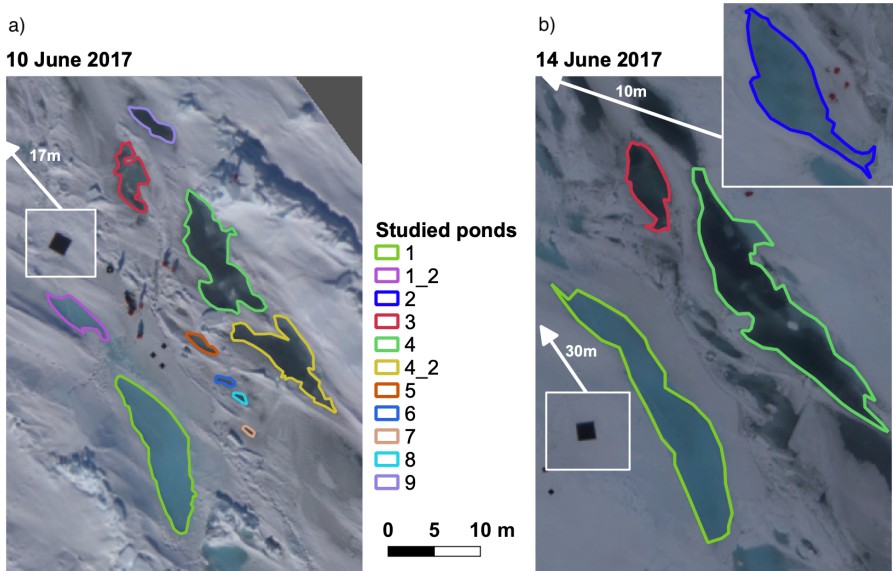

**Figure 1.** Overview orthomosaics of the PS106/1 (PASCAL) study area north of Svalbard at 81°50'North and 10°20'East in June 2017. Marked ponds were examined in this study. White inlets show the black reference targets enlarged. The known size of the targets was used for a scale check of the orthomosaics. The larger inlet in (b) shows pond #2 used only in the 14 June 2017 evaluation. White arrows connected to the inlets indicate their position in the study site, and meters give the distance from the edge of the orthomosaics. The maps are projected in UTM32N.

## 2  Overview of data and ice conditions

### 2.1  Images and field data used in the method development from PASCAL 2017

We developed and initially tested the method on data from RV *Polarstern* Cruise PS106/1 (PASCAL) that took place in June 2017 (Macke and Flores, 2018), and subsequently confirmed it for deeper ponds (>0.4 m) with data from the MOSAiC expedition. During PASCAL, we collected aerial images together with coordinated manual ground-truth measurements. We measured in situ pond depths as a reference for the development of remote sensing-based pond depth retrievals (including: König and Oppelt, 2020; König et al., 2020) and as part of a geodetic survey of ponds, including their depth and bottom ice thickness. Coordinated data to study the photogrammetrical pond depth retrieval was available from two days: 10 June 2017 and 14 June 2017. On both days, we collected RGB images above the in situ measurement area in a flight pattern that allows for the photogrammetrical retrieval of surface topography. In addition to a high overlap of images in the forward direction, which was present in nearly all images captured during the campaign, a crucial lateral offset of the acquisition positions was achieved on both days. The ice floe to which RV *Polarstern* was anchored during the campaign was located north of Svalbard at 81°50'North and 10°20'East. Given the location, we used zone 32N in the Universal Transverse Mercator system (UTM32N, EPSG:32632) as a projected coordinate system for all geospatial data evaluation.

The study area was located approximately one kilometer behind the stern of RV *Polarstern* in a younger, first and second-year ice region that was, before our visit in June 2017, subject to strong deformation (König et al., 2020). Several depressions had been formed between rafted ice floes along a ridge and were flooded with seawater. At the time of our arrival, the PASCAL floe was already subject to melting, but visible melt ponds on level ice had not yet formed. The studied depressions were thus most likely initially formed by flooding and provided, at the time of our stay at the floe, a sink for incoming solar radiation, which then led to a catalyzed melt pond formation in the particular region. Due to the diverse appearance of the underlying ice, from bright blue to almost black pond bottoms, this was an ideal study area (Fig. 1).

Images on 10 June (Fig. 1a) were acquired from RV *Polarstern* helicopter D-HARK with the implemented CANON EOS 1D Mark III 14mm lens nadir camera system during the measurement flight 20170610-2, which took advantage of the thoroughly clear sky conditions and aimed at an up-scaling of ground measurements. Therefore, several flight legs were flown in different flight levels (60 m to 3000 m) above the study area. For this method development study, relying on high-resolution data, all images were used that have been captured below 200 m flight altitude and therefore provide a ground sampling distance (GSD) more precise than 10 cm, which is at least one order of magnitude smaller than typical pond extents at the study site.

Weather conditions on 14 June (Fig. 1b) were the opposite. The entire sky was covered by a stratiform cloud cover, as is usual in central Arctic summers (Cotton et al., 2011) when the average cloud coverage reaches its maximum of about 70% (e.g., Wang and Key, 2005). No solar disc was visible, meaning incident light can largely be assumed to be diffuse at that time. Images from that day were captured during measurement flight 20170614-1. Due to the sky conditions, flight altitude was limited to 300 ft ($\approx$100 m), so no further altitude filtering was needed.

The measured ponds were assigned numbers #1 to #9 (Fig. 1). Ponds #1_2 and #4_2 merged with two larger ponds #1 and #4 over the course of four days. The selection of ponds for the analysis depended solely on the availability of in situ depth measurements carried out on-site as part of the measurement program and their location in the center of the flight pattern. Due to their position outside the photographed area, some ponds examined in König et al. (2020) had to be left out of this study.

Manual pond depth measurements were collected with a meterstick, and the locations of the measurements used here were determined relative to reference points beside the ponds or marked manually in aerial images from the previous day. We assume a horizontal accuracy of the measurement location of 0.3 m, which should not strongly impact the depth measurement accuracy in the center of ponds.

Pond depth measurements in the study area of PASCAL ranged from 7 cm to 26 cm on June 10 and 5 cm to 37.5 cm on June 14 (Table 1). We separated ponds by their either bright-blueish or dark-greyish appearance to prove the independence of our approach from the optical properties of the pond bottom. Manual depth measurements show that there were no systematic differences in depth between these groups. Drilling through the pond bottom after completing all other measurements revealed that all ponds were in an equilibrium state with sea level. Ponds #1, #3, and #4 were measured on both days, while ponds #5 to #9 were only measured on the first day. Ponds #1_2 and #4_2 had merged with #1 and #4 in the ongoing melt process. Pond #2 was measured only on the second day. Pond coverage increased notably in the study area during the four days.

**Table 1.** In situ pond depth measurement statistics and pond color type bright-blueish (bb) or dark-greyish (dg) from the PASCAL pond study site.

| # pond | number of measurements | mean depth [cm] | standard deviation [cm] | min. [cm] | max. [cm] | kind |
|--------|------------------------|-----------------|-------------------------|-----------|-----------|------|
| 10 June 2017 | | | | | | |
| 1 | 15 | 20.5 | 4.9 | 9.5 | 25.0 | bb |
| 3 | 2 | 8.5 | - | 7.0 | 10.0 | dg |
| 4 | 7 | 17.7 | 1.6 | 16.0 | 21.0 | dg |
| 5 | 1 | 12.0 | - | - | - | dg |
| 6 | 1 | 11.0 | - | - | - | dg |
| 7 | 1 | 10.0 | - | - | - | dg |
| 8 | 1 | 12.0 | - | - | - | dg |
| 9 | 1 | 9.0 | - | - | - | dg |
| 1_2 | 3 | 15.5 | 7.8 | 7.5 | 26.0 | bb |
| 4_2 | 6 | 14.4 | 2.3 | 10.0 | 17.0 | dg |
| 14 June 2017 | | | | | | |
| 1 | 3 | 22.0 | 12.0 | 5.0 | 31.0 | bb |
| 2 | 3 | 30.8 | 9.1 | 18.0 | 37.5 | bb |
| 3 | 4 | 13.5 | 1.7 | 12.0 | 16.0 | dg |
| 4 | 4 | 19.5 | 6.8 | 8.0 | 25.0 | dg |

## 2.2 Aerial image data collected during MOSAiC 2020

Aerial image collection was part of helicopter grid surveys being part of the regular measurement program executed during
the year-long drift campaign MOSAiC onboard the RV *Polarstern* from 2019 to 2020 (Nicolaus et al., 2022). During the drift, RV *Polarstern* had to be re-positioned several times, with a prolonged break in data during the initial pond formation period between 16 May 2020 and 19 June 2020 because of an inevitable crew exchange on Svalbard. At the time of RV *Polarstern*'s return for the expedition leg 4, a distinct floe had emerged, round-shaped and with a diameter of 1 km. It became the new location of the central observatory (called in Webster et al. (2022): Central Observatory 2, CO2). This floe of leg 4 formed
from a former ice formation called *the Fortress* (von Albedyll et al., 2022); strongly compressed and deformed ice (second-year and multi-year ice, SYI and MYI) adjacent to the previous legs' Central Observatory area and surrounded by some FYI areas.

     Orthomosaic and DEMs of the entire MOSAiC leg 4 floe are available from 30 June 2020, 17 July 2020 and 22 July 2020 (Neckel et al., 2022). They were compiled using the methods described in Neckel et al. (2023) and Fuchs (2023c) and cropped here to the extent of 2 km x 2 km around the floe. We classified the sea-ice surface into three main surface types (ice/snow,
ponds and open water) applying the sea ice image classification tool PASTA-ice (Fuchs, 2023c) on the brightness corrected orthomosaics (*l2* data) on 17 July 2020 and 22 July 2020 and on the brightness corrected orthomosaic with cloud correction

(*l2b* data) on 30 June 2020. The classification algorithm yields surface class maps in geospatial raster and vector data format, facilitating subsequent processing and analysis.

Photogrammetrically reconstructed DEMs from 30 June 2020 and 22 July 2020 had a vertical offset from zero caused in their processing. We leveled the open water level to zero using a flat plane fitted through all lateral snow/ice–open water boundaries positions in the DEM within the cropped extent of 2 km x 2 km. These reference points were automatically extracted from the raster data DEM at the positions of touching surface class vector polygons. For 17 July 2020, this processing was not possible, as the DEM shows substantial deviations outside of the floe area because of the non-uniform drift of surrounding smaller floes during the grid survey, making it impossible to extract the water level from the DEM. However, manual inspection of level ice areas and single well-reconstructed water–ice boundaries confirmed that no further correction was required on this day in the MOSAiC floe area.

The floe edge was retraced manually in QGIS (QGIS Development Team, 2020) to retrieve statistical data only within the MOSAiC floe area.

## 3    Method development

The method development was performed with the PASCAL data due to the availability of ground truth measurements. Flight patterns not yet adapted for depth determination also made it possible to develop a series of corrective measures, which, depending on the quality of the data, can also be used in future campaigns. All aerial images of a survey flight were taken with constant exposure settings and a mechanically and electrically fixed autofocus that was set to the flight altitude during pre-flight preparation.

### 3.1    Photogrammetric surface reconstruction

We use the commercial photogrammetry suite Agisoft Metashape to calibrate the camera optics and solve the complex aerial triangulation equations to calculate orthomosaics and DEM as georeferenced raster data. The continuous drift of the ice during the measurement was thereby automatically corrected in the bundle-block-adjustment by recalculating acquisition positions relative to the ice floe. Each surface point in the area of the studied ponds was captured on both days with more than nine different images. The ground sampling distance of both raster maps, orthomosaic and DEM, is 10 cm per pixel in the horizontal plane. The vertical resolution of the calculated DEM is $10 \times 10^{-6}$ m. Camera positions determined by aerial triangulation led to a reprojection error of images of 0.96 pixels (10 June 2017) and 1.03 pixels (14 June 2017). The ice drift was, therefore, successfully corrected by the determination of artificial image recording positions relative to the ice moved with the drift in a Lagrangian approach. Since ground control points (GCPs) with well-known position data were not available for a further accuracy assessment, a 2.00 m × 2.00 m black reference target located at the ice surface close to the ponds (Fig. 1) was used as scaling reference. Length scale accuracy of the raster data was thereby determined to ±2 %. DEMs were smoothed using the *moderate* depth map filter in Agisoft Metashape to avoid unnatural spikes in the model due to incorrect triangulation of single points.

## 3.2 Light refraction at the water–air interface

The photogrammetrical determination of the DEM relies on colinearity. Hence, optical beam paths between the observed ground surface and the lens are assumed to be straight. They are defined by the external orientation of the camera and the ground elevation. Distortions of the linear beam path are only considered in the camera optics. They are corrected with the Brown camera model integrated into the workflow. However, this basic assumption, valid for typical one-medium, low-level airborne observations, is invalidated in the case of underwater pond bottom observations by light refraction at the water–air interface. Due to the reduced speed of light in water compared to air, the electromagnetic wave, respectively light beam, is refracted more strongly away from the normal as it exits the water. The change from the angle of incidence $\beta$ to the angle of emergence $\alpha$ at the interface between water and air is described by Snell's law:

$$n_{air} \cdot sin(\alpha) = n_{water} \cdot sin(\beta) \tag{1}$$

With $n$ representing the refractive indices of air and water. Both are assumed constant in this work with $n_{air} = 1$ and $n_{water} = 1.335$ (Millard and Seaver, 1990) (value for freshwater at 0 °C increased by 0.001 to account for salt remnants in the pond water). We found that dispersion, the wavelength dependency of the refractive index, does not have to be taken into account here in the wavelength range of the camera between 300 nm to 700 nm as it leads to deviations below the measurement resolution and accuracy. Further assumptions to describe the recorded beam paths from the pond bottom to the camera are:

1. no reflection or scattering of incident light at the pond surface or in the water column

2. reflection of incident light at the pond bottom can be approximated by Lambert's law

These assumptions align with those made in Malinka et al. (2018) for the optical properties of pond bottoms with slight modifications. To match assumption 1, in clear sky conditions, all images with sun glint on the pond surface need to be removed from the analysis. However, given the typically low solar elevations in the high latitudes, there was no sun glint on ponds in the observations used here due to (i) primarily wave-free pond surfaces and (ii) almost nadir measurements. The reflection of diffuse light at the pond surfaces in overcast conditions does not affect the structure recognition. It, therefore, does not pose a problem as it does in the pure optical retrieval algorithm of König et al. (2020).

Refraction at the pond–air interface may result in underestimating pond depth in photogrammetric measurements. In the following, we follow an idealized sketch of optical paths given in Fig. 2 to describe correction factors retrieved to cope with the impact of refraction. The overarching goal of the method was to preserve or restore colinearity in the multi-view surface reconstruction so that an integrated evaluation with Agisoft Metashape is still feasible. This approach, therefore, differs strongly from previous studies, which were set up on a completely new and complex set of equations (e.g., Westaway et al., 2001), which then take into account the refraction of light but do not rely on highly specialized programs to solve it efficiently, or merely neglected the impact of optical effects (e.g., Casella et al., 2017; Divine et al., 2016).

Figure 2 shows the virtual and actual intersection of beamlines in a pond from two opposite and monocular observational positions. The actual intersection point $AP$ represents the true pond bottom, while the virtual point $TP$ is located at the depth

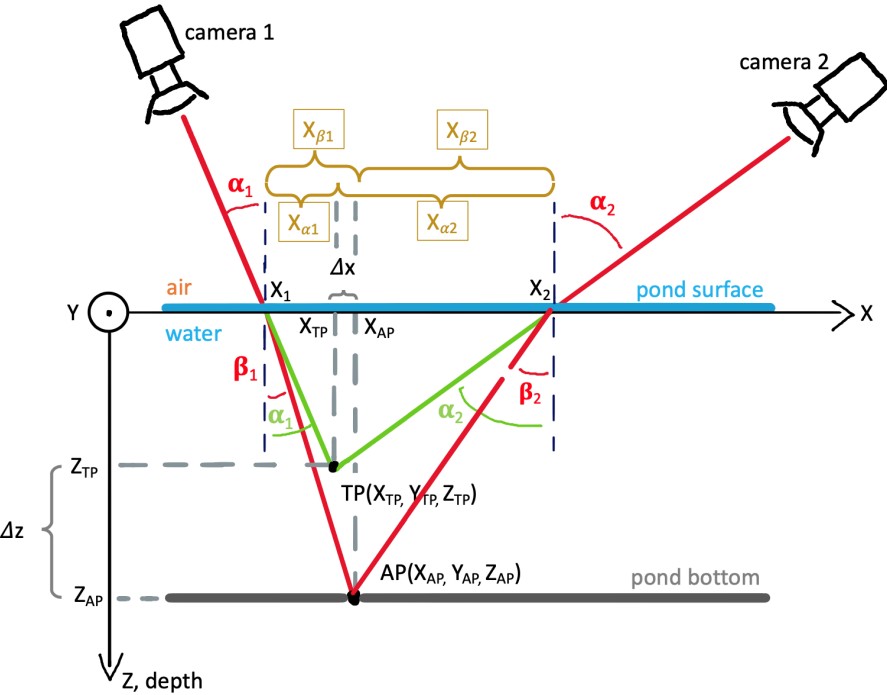

**Figure 2.** Illustration of virtual and actual intersection lines in the photogrammetric reconstruction of melt ponds. Camera 1 and 2 symbolize two individual pixels in images from different locations that captured the same point in a pond on the ice surface from a measurement angle $\alpha_1$ and $\alpha_2$. $TP$ indicates the virtual point that is photogrammetrically reconstructed from the intersecting beamlines, assuming colinearity. $AP$ indicates its actual position at the actual depth of the pond when refraction at the air–water interface is taken into account (change from angle of incidence $\beta$ to the angle of emergence $\alpha$). $\Delta x$ and $\Delta z$ quantify the horizontal and vertical deviation between both, for which correction methods are introduced in the text.

where the beamlines would intersect without refraction. The latter is the one determined by Agisoft Metashape.

To better understand the approximations derived from it, we retrace the optical path from the pond bottom to the camera in three steps: (I) A recognizable pattern allows a point on the pond bottom to be clearly identified on different images captured from different positions. We call this key point at its actual position AP and assign the Cartesian coordinates $AP(X_{AP}, Y_{AP}, Z_{AP})$.

(II) As we consider $AP$ a Lambertian reflector, identical beams radiate from the point in all unobscured directions. Two of them reach the monocular observation points camera 1 and camera 2, which symbolize two pixels in different aerial images taken along the flight track. These beamlines are not straight but bent at the water–ice interface. The angle of emergence $\alpha$ is larger than the angle of incidence $\beta$ and defined by Snell's Law: $\alpha = arcsin(n_{water} \cdot sin(\beta))$ (equation 1 rearranged). (III) The coordinates of the origin of AP are determined by aerial triangulation from all available camera positions. Since this is based on

the assumption of colinearity, the obtained virtual position of the point in the reconstruction is located in $TP(X_{TP}, Y_{TP}, Z_{TP})$. $TP$ deviates in height from the original position $Z_{AP}$ by $\Delta z$ and since camera positions usually do not all have precisely the same elevation angle also in its horizontal position $X_{TP}$ and $Y_{TP}$ by $\Delta x$.

This results in two deviations in the colinearity approximation caused by the pond water that must be corrected or avoided, a vertical and a horizontal one. Both deviations are discussed separately in the following two paragraphs.

### 3.2.1 Horizontal deviation

The horizontal deviation $\Delta x$ potentially causes a mismatch of point detections and should, therefore, be avoided in an integrated scheme. We consider it sufficiently suppressed when $\Delta x$ is smaller than the ground sampling distance. $\Delta x$ is directly dependent on both angles of emergence and the measured pond depth $Z_{TP}$. We define the ratio between horizontal deviation and measured pond depth as deviation factor *specific horizontal mismatch* $\kappa$, with:

$$\kappa = \frac{\Delta x}{Z_{TP}} \tag{2}$$

Figure 3 shows how $\kappa$ changes with both incident angles. In a flight altitude of 300 ft, which is a good choice for highly resolved pond studies, the ground sampling distance of the CANON camera system is approximately 0.05 m. This means for all measured pond depths $Z_{TP}$ up to 1.5 m, the maximal horizontal deviation $\Delta x = \kappa \cdot Z_{TP} = 0.042$ m caused by refraction remains below the measurement resolution and the photogrammetric projection error when we restrict incident angles to $\alpha_{max} = 40°$. We see it as a good compromise between moderate horizontal deviation and enough field-of-view to preserve sufficient overlap of images. Therefore, all image pixels at larger opening angles relative to the nadir are neglected in the reconstruction. This is done by creating masks individually for every single image depending on the orientation angle of the camera derived from the camera alignment process in Metashape.

### 3.2.2 Vertical correction

After examining how horizontal deviations can be avoided, this section concerns the underestimation of the measured pond depth $Z_{TP}$ owing to refraction. We discuss how it can be corrected with a correction factor $\gamma$ defined by $Z_{AP} = \gamma \cdot Z_{TP}$. $\gamma(\alpha)$ is given by:

$$\gamma(\alpha) = cos(\alpha) \cdot \sqrt{n_{water}^2 - sin^2(\alpha)} \tag{3}$$

It can be shown that the correction factor $\gamma$ converges towards the refractive index of water $n_{water}$ for arbitrarily small $\alpha$ and eventually becomes equal to $n_{water}$ at exactly $\alpha = 0$ (see supplementary material for the equation solutions).

$$\lim_{\alpha \to 0} \gamma(\alpha) = n_{water} \tag{4}$$

This contradicts the statement in Casella et al. (2017) that refraction does not influence underwater depths retrieved from almost nadir images. Instead, our mathematical and geometrical evaluation shows that almost nadir depth measurements must be multiplied with the refraction index to achieve correct depth values. When $\alpha$ increases, $\gamma$ rises slowly at first and eventually very strongly (Fig. 4a). In the previously defined range of maximal 40° incident angle, $\gamma$ reaches a maximum value of $\gamma_{max} = 1.527$. However, since derived depths result from averaging numerous intersection rays with mostly small angles of incidence, the

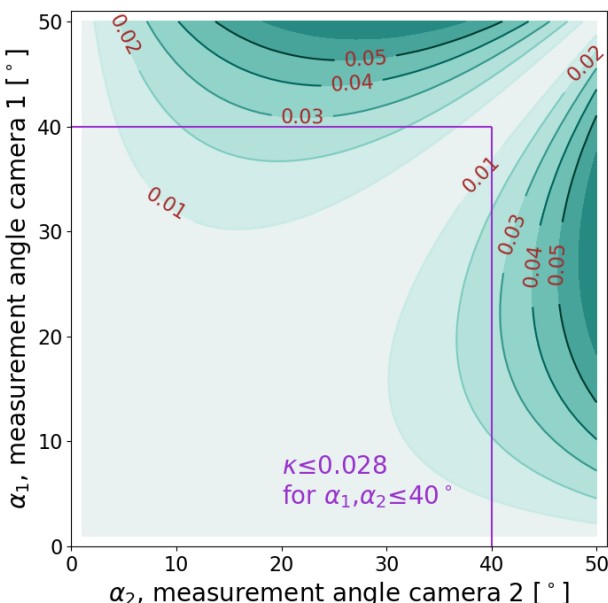

**Figure 3.** Geometrical evaluation of equation 2 (including equation A6), showing how the *specific horizontal mismatch* $\kappa$ changes with two measurement angles $\alpha_1$ and $\alpha_2$. We found that below $40°$ measurement angle errors induced by the horizontal mismatch remain small enough to be neglected in the automatized reconstruction.

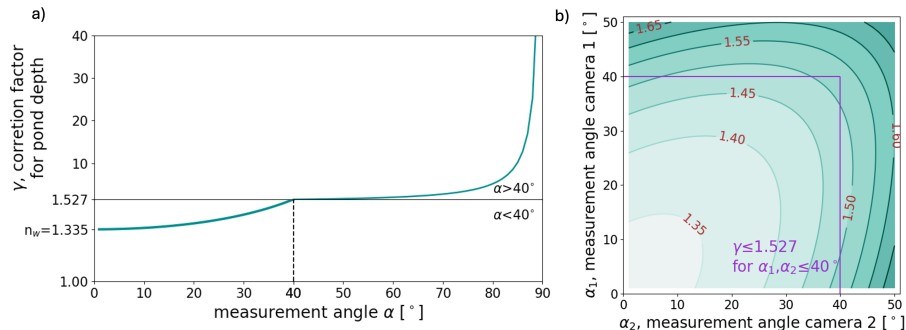

**Figure 4.** Geometrical evaluation of depth correction factor $\gamma$ for equal measuring angles (a) and differing measuring angles (b). $n_w$ is the refractive index of water. The vertical axis of panel (a) is split at $\gamma = 1.527$, which equals the depth correction factor for the limited measurement angle $\alpha = 40°$ (indicated by the black lines). $\gamma$ strongly increases for $\alpha > 40°$. In panel (b), the $40°$ threshold is marked with the purple lines.

small increase of $\gamma$ is ignored, and $\gamma$ is kept constant and equal to the refractive index of water $n_w$ in our method for the correction of all underwater pixels.

An additional conclusion can be drawn for the horizontal deviation from this analysis. Figure 4 shows that water depths are systematically underestimated with the measured pond depth, which means that the previously obtained maximum opening

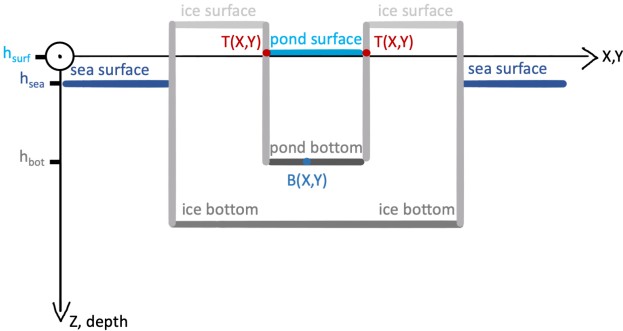

**Figure 5.** Illustration of different elevation and depth levels relevant for the determination of individual pond depths. Schematically, $h_{surf}$ is the vertical position of a pond surface, $h_{bot}$ is the vertical position of its pond bottom, and $h_{sea}$ is the vertical position of the sea surface. The bathymetric map B(X,Y) describes the actual pond depth at the projected geographic North and East coordinates (Y and X) in the in reality not uniformly deep ponds. T(X,Y) is the height of the pond margin at coordinates X and Y.

angle $\alpha_{max}$ in fact allows for greater actual pond depths than assumed in section 3.2.1, in which we limited the virtual pond depth $Z_{TP}$ to 1.5 m.

### 3.3 Pond depth determination

Pond depth $d$ is the vertical extent of the water column in ponds. It is composed of the vertical position of the pond bottom $h_{bot}$ and the height of the pond water surface $h_{surf}$ (Fig. 5). Large altitude inaccuracies of aircraft positioning systems, especially in high latitudes, make it impossible to use the GPS aircraft altitude and the reconstructed distance to the ground as an absolute height reference above sea level. That is why studies of ice topography with, for example, ALS, usually use areas of very thin ice or open water to be referenced to water level (e.g., Hutter et al., 2023). For ponds, the reference is even more complex

since pond depth is, as previously mentioned, not only prescribed by the topography of the pond bottom but also by the individual height of the water level in the pond. This water level is typically above sea level, partially caused by impermeable sea ice or later in the season, when ice is typically permeable, by the density difference between freshwater in the pond and underlying seawater. The pond water level is individual for every single pond, especially during the early stages of melt pond formation, i.e., the time in melt season before the ice gets permeable enough to allow for pond drainage (cf. stage I, Eicken

et al., 2002). Hence, we calculated relative pond depth without an absolute reference. To this end, the water level in each pond was determined by its height in the DEM at the edge of the pond. The method, therefore, highly benefits from the optical aerial images enabling high-resolution surface type classification with PASTA-ice (described in the next section) on exactly the same raster as the reconstructed topography data. We derived the pond margins as vectors (linestrings) from the classified images and overlaid the photogrammetrically retrieved DEM raster data $T(X, Y)$ with the pond margins. Then, we extracted the relative

height of the pond surface $h_{surf}$ at the pond margin from the DEM, where this marks the transition from ice to pond in the smoothed topography. Due to the smoothing, we expect the method to work also for ponds with almost vertical walls later in the season (Fetterer and Untersteiner, 1998), which were not part of the evaluation set, however. This extraction was done using

**Table 2.** Pixelwise input features to classification scheme

| Feature | Equation | Reference |
|---------|----------|-----------|
| Red | $R_{8bit}$ | Standard |
| Green | $G_{8bit}$ | Standard |
| Blue | $B_{8bit}$ | Standard |
| BR1 | $G-R/G+R$ | (Wright and Polashenski, 2018) |
| BR2 | $B-R/B+R$ | (Miao et al., 2015) |
| BR3 | $B-G/B+G$ | (Miao et al., 2015) |
| BR4 | $G-R/2B-G-R$ | (Miao et al., 2015) |
| BR5 | $B+G-2R$ | New |

the Python libraries geopandas, rasterio, and rasterstats (Jordahl et al., 2020; Perry, 2015; Gillies, 2013, respectively). The two-dimensional bathymetric map $B(X,Y)_i$ of each individual pond $i$, with $X$ and $Y$ as geographic North and East coordinates in the projected coordinate system, was subsequently retrieved from:

$$B(X,Y)_i = -(h_{surf_i} - T(X,Y)) \cdot n_{water} \tag{5}$$

where $n_{water}$ is the refractive index of water as depth correction factor as discussed in the previous section 3.2.2. Depth is specified positive down. Last, we compiled a pond-depth corrected topography map $T_{pond}(X,Y)$ from the individual pond bathymetries.

## 3.4 Pond classification with PASTA-ice

For the automatic detection of ponds in aerial images, we used the Proportional Analysis tool for Surface Types in Arctic sea Ice images (PASTA-ice, source code repository in Fuchs (2023a), previously used in Thielke et al., 2023; Niehaus et al., 2023). In the following, we briefly summarize the algorithm due to its importance for workflow automation. Details, including the method development and evaluation, are found in Fuchs (2023c).

PASTA-ice is tailored to the aerial images captured with the AWI imaging system. Besides focusing on the semantic separation into surface type classes, the algorithm aimed at retracing pond outlines used to extract pond levels and to compile statistics on individual ponds. Image classification is done pixel-wise in brightness-corrected orthomosaics (e.g., Neckel et al., 2023) based on absolute R, G, B values and ratios thereof (Table 2).

Classification is performed with the random forest classifier implementation in Scikit-learn (Pedregosa et al., 2011). The classifier was trained and tested using data from manually selected areas in very diverse sea ice surface appearances recorded during PASCAL. Pixels in orthomosaics are classified into nine different sea ice surface sub-classes that belong to three main classes (Table 3): snow/ice, open water, and ponds (including submerged ice). Adjacent sub-class pixels of similar main classes are subsequently combined into main class vector objects if these consist of, at minimum, 100 pixels (the threshold was chosen

**Table 3.** Sea ice surface type classes used in the classification tool PASTA-ice to semantically segment orthomosaics.

| Main surface type class | sub-classes | notes |
|---|---|---|
| open water | open water | |
| snow/ice | snow / white ice | |
| | bare/wet ice (greyish) | |
| | bare/wet ice (blueish) | |
| | shadow on snow/ice | |
| ponds | bright blue ponds | |
| | dark/grey ponds | |
| | shadow in pond | |
| | submerged ice | pure optical classification, not location dependent |
| submerged ice | all pond subclasses | pond objects located between snow/ice and large open water areas (post-processing) |

similar to Huang et al., 2016). This threshold is applied as a minimum area requirement to match the baseline of high-resolution
data that objects are resolved from various pixels (e.g., Wright and Polashenski, 2018). The chosen threshold corresponds in the
orthomosaics of this study to an area of approximately 1 m$^2$ (PASCAL) and 25 m$^2$ (MOSAiC). Smaller objects are considered
noise and are added to the largest adjacent object; their area fraction is taken into account when estimating the inaccuracy of
the classification result.

For spatial analysis, main class objects are converted to polygon geometries defining the outer and, if present, inner edges
of the object. They also include an attribute table that contains information on the sub-class proportions and a classification
confidence proxy from the prediction probability output of the classifier (Fuchs, 2023c). Classification recall and precision are
somewhat limited for the very specific sea-ice surface subclasses listed in Table 3 due to overlaps in appearance but high for
the combined main classes (Fig. 6). High accuracy values mainly result from large sample sizes, resulting in large numbers
of true negative pixels. All pond objects are reclassified to submerged ice if they are located spatially between a snow/ice and
open water object and if the open water object is larger than the pond object.

The polygons of ponds, i.e., their outlines defined by vertices and connecting edges that resemble the snow/ice to pond
interface, are used to extract the pond level from the DEM. The very high accuracy in classifying the main classes indicates a
sufficient detection of the transition areas from pond to ice and, thus, of the pond polygons.

### 3.5 Pond margins detection and height correction in the PASCAL data.

Pond margins in the PASCAL study area were traced manually in orthomosaics using QGIS (QGIS Development Team, 2020)
to better asses the depth correction algorithm without the impact of any classification inaccuracies expected in this particular
study area (traced polygons shown in the study site overview, Fig. 1). Due to the deformed ice surrounding this specific

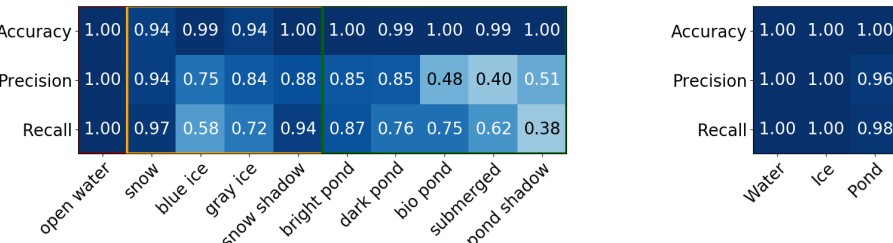

**Figure 6.** Accuracy, recall, and precision evaluation for sub- and main surface type classes in the PASTA-ice classification scheme retrieved from a test data set compiled from images collected during PASCAL. The orange and green lines mark the separation between open water and ice classes and between ice and pond classes, respectively. Detail of Fig. 3.19 (c, f) in Fuchs (2023c).

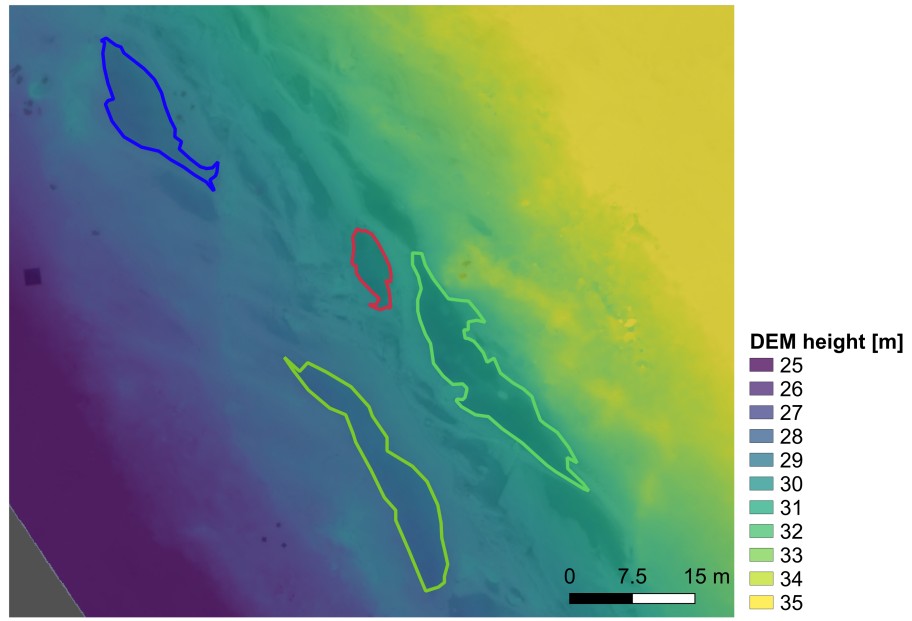

**Figure 7.** Photogrammetrically reconstructed DEM of the study site on 14 June 2017 with marked ponds (#1-#4 from Fig. 1). Missing GCPs led to a strong tilt of the surface. Inaccurate GPS heights and reconstruction in the WGS84 ellipsoid cause an absolute offset from 0 m.

location, shadows tended to impact the automatic classification scheme in PASTA-ice on this small scale, which eventually could strongly falsify the pond exterior detection needed to derive $h_{surf}$. Especially since misclassified shadows that stretch from ponds into adjacent ridges can be partially far above the water level. In larger sample sizes and more even ice areas, where most ponds usually form during summer melt, automatic surface classification with pond margins detection is easily possible, as shown in Section 5 with the MOSAiC data.

Missing GCPs and slightly arbitrary camera optics mainly caused by a bubble-shaped protection window in front of the moveable mounted camera - for shock protection - furthermore led to large-scale deviations like curved and tilted surfaces in

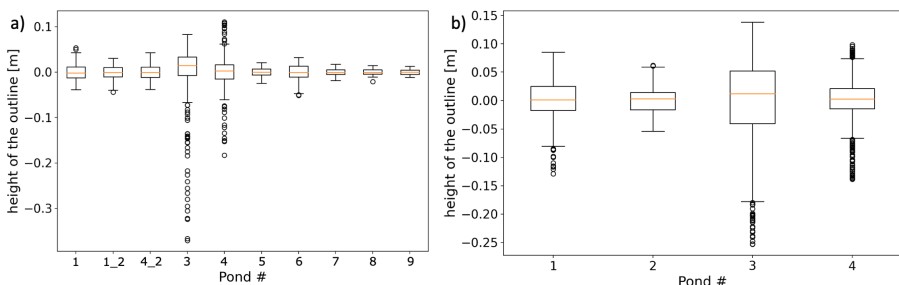

**Figure 8.** Deviation of the pond outline topography given by T(X,Y) to the fitted pond surface planes $h_{surf_i}$ on 10 June 2017 (a) and 14 June 2017 (b).

the Agisoft DEM retrieval of the PASCAL data (Fig. 7). These deviations were approximated as a linear slope on the length scales of ponds. As a correction, we fitted a two-dimensional plane through each pond outline in the DEM. This regressed plane is assumed to represent the water level $h_{surf_i}$ in equation 5.

$$h_{surf_i}(X,Y) = a \cdot X + b \cdot Y + Z \tag{6}$$

$Z$ is the absolute height correction resulting from the mean vertical deviation of the retrieved surface from sea level. This mean vertical deviation is mainly caused by inaccurate GPS altitudes at higher latitudes. Height levels along the corrected pond edge deviate only slightly from zero (mostly less than ±0.05 m) as shown in the box-and-whisker plots in Fig. 8. In bending-free DEMs (e.g., the ones from MOSAiC presented below), the reference height can be derived solely from the mean elevation of the pond outline $h_{surf_i}$.

## 4   Results of the method development

### 4.1   Photogrammetrically derived pond depth on PASCAL compared to manual measurements

Bathymetric charts of ponds in the study area were calculated from aerial images for all manually sampled ponds as previously described (Fig. 9). To account for the location inaccuracy of the in situ data, photogrammetrically derived pond depth data were averaged in a circle with a radius of 0.3 m around the point measurements (Fig. 9).

In situ and photogrammetry yielded almost identical pond depths (Fig. 10). The original pond color (see Fig. 1) indicated by the color of the dots (blueish, bright, or greyish, dark ponds) in Fig. 10 does not affect the reconstruction. However, pond size shown by the size of the dots in Fig. 10 seems to have one specific influence on the reconstruction: small ponds (<1 m in diameter), indicated by tiny dots, are underestimated in their pond depth. As bias of the measurement method, we retrieve BIAS=-1.2×10$^{-3}$ m with a root mean square error of RMSE=3.84×10$^{-2}$ m and mean absolute error of MAE=2.65×10$^{-2}$ m.

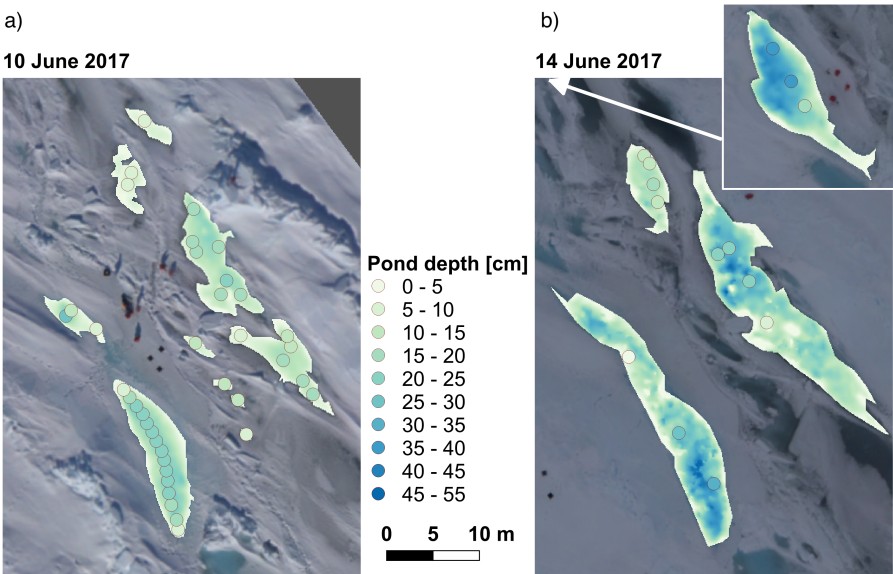

**Figure 9.** Photogrammetrically derived pond bathymetry charts of all ponds in the PASCAL study area, for which in situ data (dots) were available. The colored shapes show the photogrammetrically reconstructed pond bathymetries and the dots show the in situ measurements. The size of the dots equals the area used for averaging the photogrammetrically derived depths before comparing them to the in situ data.

## 4.2 Photogrammetrically derived pond depth on MOSAiC compared to echosounding data

It was noted above that ridge structures directly adjacent to the ponds in the PASCAL study area required a manual tracing of pond polygons to get a precise reference water level. Improved flight patterns during the MOSAiC expedition (mowing-the-lawn pattern) with a regular lateral and forward overlap in images, together with well-classifiable surface conditions, increased the capability of the algorithm to work entirely autonomously. We evaluate the photogrammetrically derived pond depths with the newly developed autonomous pond investigation system *Böötle*, equipped with a downward-facing echo sounder (Oppelt

and Linhardt, 2023). We compare pond depths of a lake-like pond *Mystery lake* which reached depths of more than 2.5 m. It was regularly mapped during helicopter survey flights and with the *Böötle*. We chose the two datasets collected closest in time on 7 July 2020 (helicopter survey) and 9 July 2020 (echosounder measurements). However, the temporal difference of two days restricts us from retrieving precise errors and corrections, but we can still use it to confirm the overall method. The comparison is particularly interesting because the pond exceeded the maximum depths assumed in the method development, and due to

smaller lateral image overlap, the opening angle in the acquired images could not be limited to 40°. Figure 11 compares the photogrammetrically derived and echosounding pond depths. Overall, both methods agree in pond depths for *Mystery lake*. Yet, the data show a divergence at greater depths (>1 m) that can be attributed to either the further deepening of the pond within the two days, a systematic underestimation of the reconstructed depth, or both. The latter would be a consequence of

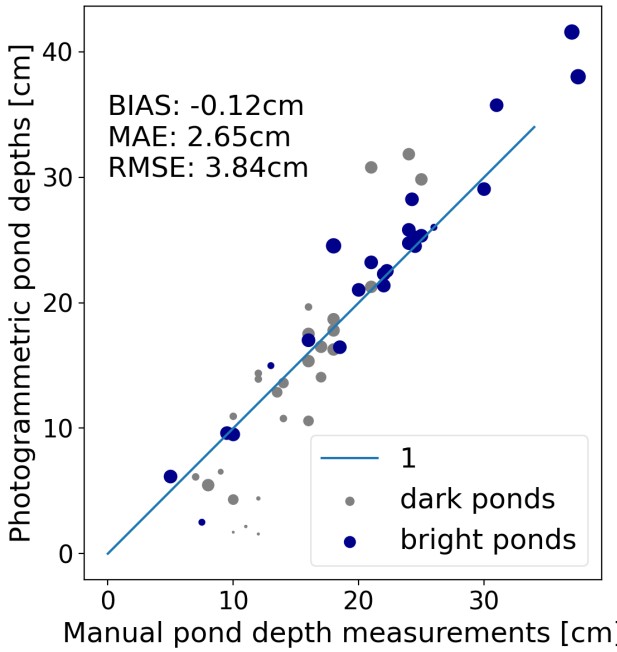

**Figure 10.** Pond depth comparison between manual in situ data and photogrammetrically derived bathymetry. The dots' colors indicate the ponds' original color (blueish, bright, or greyish, dark ponds); see Fig. 1. The size of the dots indicates melt pond size ranging from 0.5 m$^2$ to 116.9 m$^2$.

the unrestricted opening angle, which actually would require a larger correction factor than what is applied. The percentage
deviation was -11.5 % at depths below 1 m and raised to -24.2 % in greater depths.

### 4.3 Impact of flight pattern

Since mowing-the-lawn is among the most time-consuming flight patterns, we further determined to what extent other flight patterns, for example, straight flight legs with sufficient forward overlap, also lead to reasonable results. Based on pond #1 (10 June 2017), we investigated how the measurement accuracy depended on the overlap of images and the lateral offset of the
flight lines. Figure 12 shows bathymetric charts of pond #1 retrieved from (a) a few measurement positions along a straight line (4 images), (b) more measurement positions along a straight line (10 images), (c) similar measurement positions with lateral offset (9 images), (d) many measurement positions with lateral offset respectively all available measurement positions as used for the comparison with in situ observations before (31 images). The camera matrix and image recording positions were optimized before and kept constant during this study so as not to impact the error. The accuracy strongly increases
with an increasing amount of measurement positions, larger incident angles, and especially, lateral offset of the measurement positions when comparing measurement positions on one straight line or isotropically distributed over the pond (Fig. 13). A reconstruction from 10 images recorded along a straight line yields a higher error of 7.90 cm compared to 9 images with both

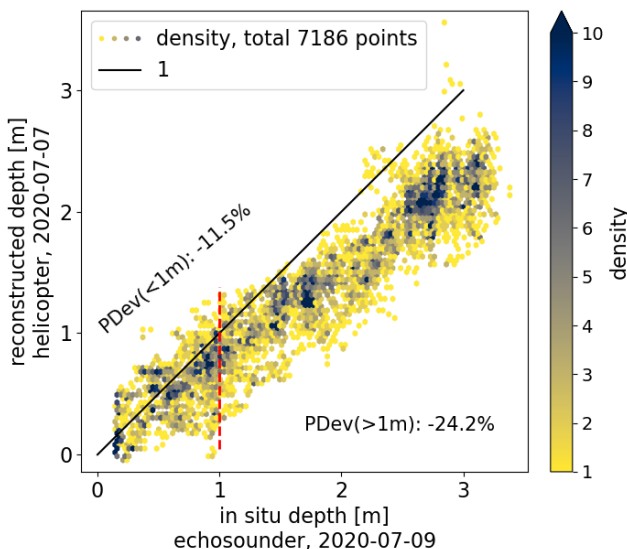

**Figure 11.** Hexagonal binning diagram showing the comparison of pond depth measurements in *Mystery lake* on MOSAiC from an in-situ echosounder (deployed on the platform *Böötle*, 2020-07-09) and photogrammetrically derived bathymetry (helicopter survey flight, 2020-07-07). A total of 7186 measurement points were compared. Only bins that contain at least one point are shown. Percentage deviations (PDevs) are calculated for echosounder depths below and above 1 m (separated by the red dashed line).

lateral and forward overlap (6.98 cm). The quantitative differences are still small, while subjectively, the differences in Fig. 13 even exceed these error estimates. Lateral offset can best be achieved with a mowing-the-lawn flight pattern.

## 5   Melt ponds on the MOSAiC floe

Regularly flown floe grids during MOSAiC (with mowing-the-lawn flight pattern) combined with the newly developed pond bathymetry retrieval enable an unprecedented three-dimensional analysis of melt ponds. Given that with the flight pattern, lateral and forward overlap in images was achieved, and owing to open water areas around the major floe and the DEM correction with ALS data by Neckel et al. (2023), a leveling of all data to sea level was possible. To do so, we fitted a flat plane through all snow/ice–open water edge positions in the DEM around the floe and subtracted it from the DEM. For the first time, we could thus retrieve pond bathymetries, pond level to sea surface height, and track pond changes from flight to flight with aerial imaging.

### 5.1   Evolution of pond coverage, bathymetry, level and volume

The first pond formation on the MOSAiC floe happened already in May 2020 as observed from satellite images (Webster et al., 2022), at the time of the absence of RV *Polarstern* from the MOSAiC floe for crew exchange. After the return in mid-June, continuous pond formation was observed. Pond coverage on the MOSAiC floe between the end of June and late July varied

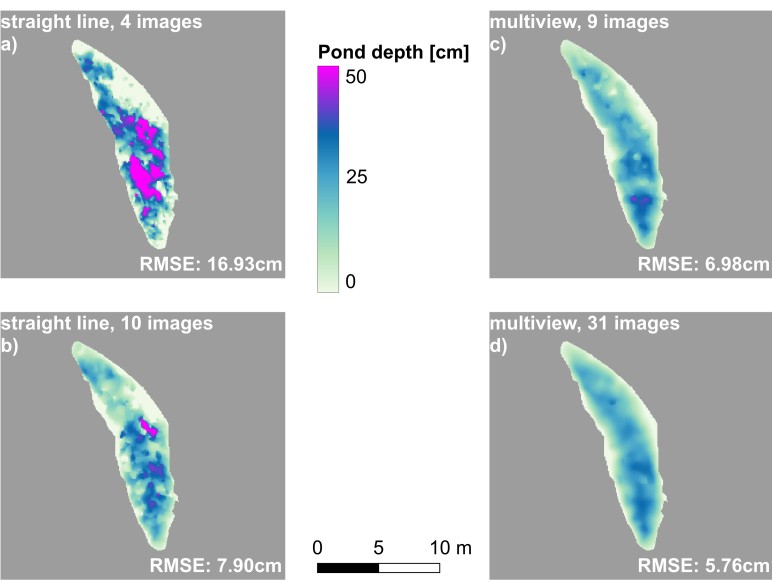

**Figure 12.** Photogrammetric pond bathymetries of pond #1 (10 June 2017) derived from different subsets of image recording positions used in the reconstruction. (a) reconstructed from 4 images along a straight line with almost nadir measurements. (b) reconstructed from 10 images along a straight line. (c) reconstructed from 9 images scattered above the pond, almost nadir measurements. (d) reconstructed from 31 images scattered all around the pond. RMSEs are calculated by comparing the reconstructions to the in situ measurements.

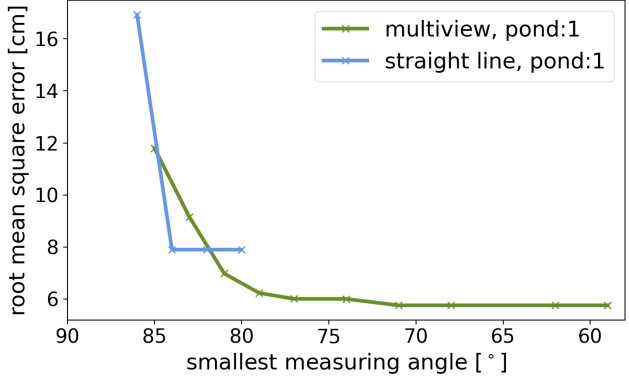

**Figure 13.** Comparison of the reconstruction accuracy using images only along a line or from horizontally distributed positions. RMSE of pond #1 depths (10 June 2017) derived from the comparison of in situ measurement points to reconstructed pond depths with the same logic as in Fig. 10. Shown are RMSEs for different subsets of image recording positions used as input to the reconstruction. Differentiation is made between points along a single line and points scattered isotropically above the pond. The smaller the smallest measurement angle, the farther away the most distant recording point is and the larger the offset.

**Table 4.** Pond coverage on the MOSAiC leg 4 floe retrieved from helicopter aerial imaging and PASTA-ice classification on three different days. Numbers in brackets depict the confidence range obtained from a newly developed confidence evaluation (section 3.4).

| Day | Pond coverage |
|---|---|
| 30 June 2020 | 22.3% [21.3%,26.0%] |
| 17 July 2020 | 22.0% [20.9%,24.3%] |
| 22 July 2020 | 23.7% [22.9%,27.6%] |

between 22.0% to 23.7% (Table 4). At the end of June, the pond cover was well developed and amounted to 22.3 % (Fig. 14). Mid-July (11-13), most ponds drained (Webster et al., 2022). Having orthomosaics of the entire floe from 30 June 2020, 17 July 2020, and 22 July 2020 (Fig. 14), we thus lack comprehensive aerial data on the days of drainage but cover both the period before and after. On 30 June, several vast, exceptionally deep ponds (>2 m) had formed on the MOSAiC floe, along with many smaller ponds (Fig. 14). After drainage, the pond cover became more fragmented and braided-like, with more shallower ponds (<1 m) in the center of the floe. We discuss this significant loss in depth of the largest ponds (incl. *Mystery lake*) in the discussion section. A direct comparison of the conditions before and after the drainage event shows that individual ponds in the strongly deformed center of the floe and smaller ponds have remained unchanged in their shape or have even grown. All prominent large ponds on the floe underwent major changes during drainage and show bare patches of ice that were formerly submerged under pond water. Nevertheless, total pond coverage was relatively constant at 22 % at the time of the three measurement flights (Table 4). A possible underestimation of the pond coverage from the PASTA-ice classification was relatively strong on 30 June 2020 and 22 July 2020, with 3.7 % and 3.9 %, respectively. This possible underestimation was presumably caused by small pond objects in the classification output that were below the 100 pixels minimum threshold for objects to be resolved, as described in the method section.

Previous studies found that pond surfaces before drainage were above sea level, forming a hydrostatic head (e.g., Perovich et al., 2021, based on SHEBA data). On the MOSAiC floe, on 30-June, two weeks before the main vertical drainage event, we find, in contrast, that 90.1% of the total pond area was already close to sea level (less than 0.2 m above sea level). Only a few small ponds (7.1% of the entire pond area) in the strongly deformed center of the floe were well above sea level (more than 0.3 m above sea level). The 50 largest ponds (covering 64.7% of the total pond area) were closer to sea level and, therefore, had a low hydrostatic head.

The partitioning of meltwater has raised particular interest on MOSAiC since distinct freshwater lenses were observed around and below the floe, strongly impacting the physical and biological sea-ice system (Smith et al., 2023). Melt ponds form a reservoir in the meltwater budget. Their bulk volume is, therefore, of great interest to better assess and understand the total budget. Having derived pond bathymetry maps of the entire floe, we can, for the first time, derive the overall volume of this meltwater reservoir directly, in contrast to extrapolating it from single transect lines, which was the only available method before (e.g., Perovich et al., 2021; Webster et al., 2022). The bulk volume of meltwater in ponds on the floe results from the

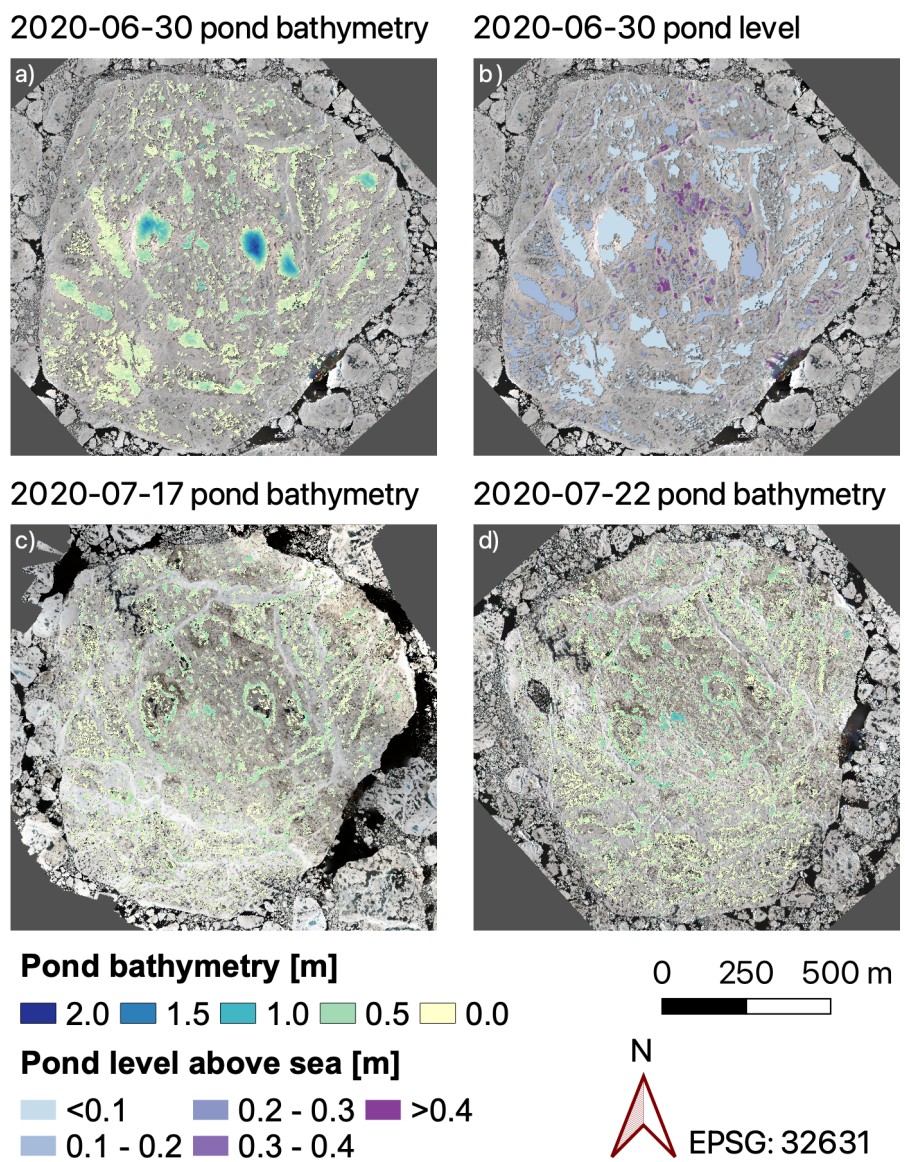

**Figure 14.** Overlays of photogrammetrically reconstructed pond data on orthomosaics of the MOSAiC floe from different MOSAiC floe grid surveys (orthomosaics and DEM from Neckel et al., 2022, reprojected in UTM31N). The panels show pond bathymetry (a) and pond level above sea surface height (b) reconstructed from an airborne survey flight on 30 June 2020, and pond bathymetries from flights on 17 July 2020 (c) and 22 July 2020 (d). Ponds were detected automatically using the PASTA-ice surface type classification.

pond bathymetry and the area covered by ponds. Deriving it as area-specific value, we find that area-specific pond volume (asPV) on the entire floe was fairly constant at 0.040 $m^3$ $m^{-2}$ to 0.051 $m^3$ $m^{-2}$ (volume meltwater per floe area, Fig. 15a).

The floe-wide, high-resolution data give us the opportunity to further investigate the spatial and temporal evolution of the pond volume. Before drainage, half of the meltwater in ponds was stored in particularly wide ponds with large diameters, quantified here by the diameter of the biggest disk that can be fitted into the pond shape. With the drainage event, such wide ponds like *Mystery lake* disappeared (Fig. B1e), and ponds with diameters 5 m to 20 m got the largest sinks for meltwater (92.5 %). Some of them were particularly large in area (up to 13000 m²) but simultaneously small in diameter (<15 m) (Fig. 15b, pond marked in *darkkhaki*). After the drainage event, only half of the meltwater (2020-07-17: 53.6 %, 2020-07-22: 52.7 %) was stored in ponds with diameter >12.5 m. Ponds of similar diameter increased in depth with time by 2.5 cm between each flight (median depth increase) and therefore compensated for the loss of volume in the particularly wide ponds that vanished with the drainage event and, with that, decreased the mean pond depth (mD) by up to 4 cm (Fig. 15c). At the same time, the overall melt pond coverage (MPF) remained relatively constant. Still, pond changed their appearance strongly to braided-like pond patterns with smaller diameters but large connected pond areas (Fig. 14). It is generally noticeable that the pond depth correlates strongly with the diameter size introduced here, with larger diameters allowing for greater depths (Fig. 15c). Following the shape of the depth $d$ to diameter $\phi$ dependency, we approximated a square root function with the least square method to the binned data (min. three ponds per bin). For the time before the drainage event, we yield $d[m] = 5.36 \cdot 10^{-2}\sqrt{\phi[m]}$. After drainage, the relation changes to $d[m] = 6.61 \cdot 10^{-2}\sqrt{\phi[m]}$. Both fits approximate the individual pond depths with a RMSE of 5.5 cm to 5.7 cm. However, the continuous deepening of the ponds after the drainage event clearly shows that this fit is limited in time. The shallower depth of smaller ponds explains why they contribute less to the total pond volume than to the total surface area of ponds, visible through the slightly skewed distribution function (Fig. 15b). We also included pond volume fraction, areal fraction, and depth, resolved over pond area in the supplementary material (Fig. B1). However, signals relative to the pond area are much less pronounced than if they are separated by the diameter measure used here.

## 5.2 Comparison to other in situ and satellite observations

Measurements on MOSAiC were carried out using many different methods and may lead to different results. In the following, we compare the aerial derived data to available results from high-resolution satellite observations (Webster et al., 2022, using the Wright and Polashenski (2018) classification algorithm OSSP) and in situ transect lines (Webster et al., 2022) to assess the accuracy of our results and the representativeness of observed areas. Transect lines were repeatedly revisited paths on which extensive in situ measurements of ice thickness, snow and ponds were carried out. The transect considered here was the longest and surrounded the entire floe (Webster et al., 2022, Fig. 2). To compare to these transects, we derived pond properties additionally within a 10 m buffer zone around retraced transect footpaths in the aerial images. Satellite and aerial-derived pond coverage of the floe is temporally sparse, but both remain relatively stable at around 22 % over the entire observational period with differences <2 %. Along the transect lines, helicopter-derived pond coverage increases from ∼10 % in late June to ∼20 % in mid-July. It thus resembles the evolution of the in situ observed coverage (Fig. 16a). Small under- and overestimations of 2 % to 5 % occur, probably caused by collocation inaccuracies. All products thus seem to be comparable for their observed

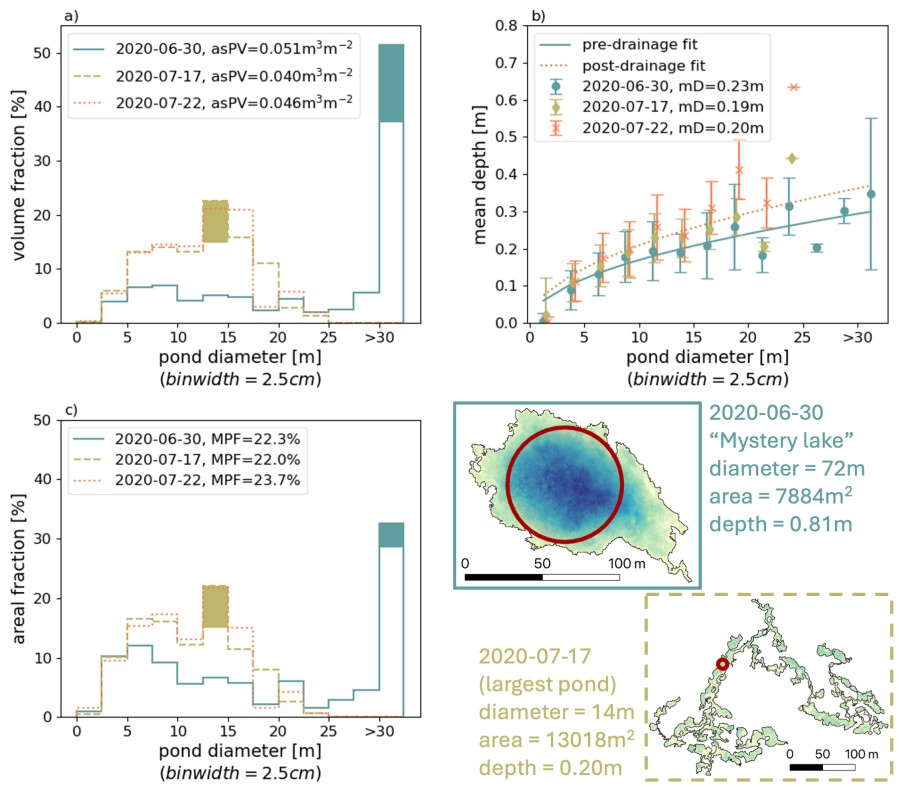

**Figure 15.** Distribution functions of pond volume (a), pond areal fraction (b), and mean pond depth (c) in 13 pond diameter size bins from 0 m to 30 m and larger than 30 m on the MOSAiC floe. Data retrieved from photogrammetric reconstructions from aerial images collected on survey flights on 2020-06-30 (pre-drainage) and 2020-07-17, 2020-07-22 (post-drainage) are shown. The total floe area-specific pond volume (asPV) is given in the legend of (a), and the total pond coverage (MPF) on the floe is in the legend of (b). Mean pond depths (mD) on the entire floe are listed in (c). Vertical error bars in (c) depict the standard deviation of pond depth in the diameter bin. For better visibility, their x-position in the graph is slightly offset. Fits show square-root functions fitted to the binned data with the least square method. Filled areas in (a) and (b) show color-coded contributions of the two ponds, whose bathymetry and shape are shown on the lower right. The colormap of the bathymetry reaches from 0 m to 2 m depth (Fig. 14). Red circles show the largest disk that can be fitted into the pond shape, which is used here to quantify pond diameters.

area. However, it becomes apparent that along the floe edge, pond coverage doubled between the end of June and mid-July and thus acted differently from the relatively constant pond coverage on the entire floe.

Pond depth along the transect line was observed about 2.5 cm higher ($\sim$15 % to 20 %) than the mean derived pond depth for the same area from the aerial derived bathymetric maps (Fig. 16b). The (very variable) pond depth on the entire floe is underestimated by the transect area in June. It matches well in July after drainage, probably because the extraordinarily deep ponds in the floe center flattened, making the pond cover more uniform. Both methods resolved a slight increase in pond depth at the floe edge. Before harmonization through the drainage event, the very deep ponds in the middle of the floe, a region

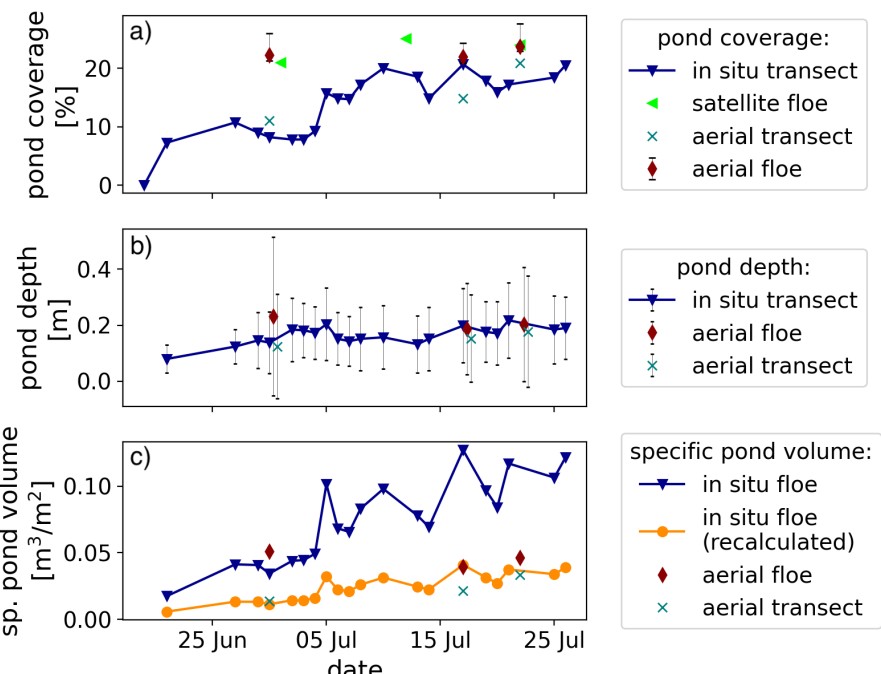

**Figure 16.** Timeseries of melt pond coverage (a), mean pond depth (b) and specific pond volume (c) retrieved from in situ transects, high-resolution satellite images (both Webster et al., 2022) and helicopter-borne aerial images. Floe properties are retrieved for (aerial and satellite) or extrapolated to (in situ) the entire MOSAiC leg 4 floe. Transect properties are measured along the transect line (in situ) or retrieved within a 10 m buffer zone around the retraced line (aerial). The in situ specific pond volume is the pond volume presented in Webster et al. (2022) divided by the floe size. However, due to a correction, we recalculated the specific pond volume using their mean pond depth and pond coverage values (in consultation with the authors). Both lines are included to avoid misunderstandings.

not covered by the transect studies, caused a more constant pond volume than extrapolated by Webster et al. (2022) from the transect area (Fig. 16c). Good agreement between both methods along the transect lines suggests that differences occur mainly due to different observed areas instead of methodical differences.

### 5.3 Upscaling factor for in situ depth measurements

In situ measurements in ponds are often restricted to very few single points. We investigate whether and how the mean pond 465 depth of entire ponds can be extrapolated from single in situ point measurements by subsampling our high-resolution photogrammetric pond bathymetry reconstruction. To this end, we benefit from the unprecedented data set in resolution within ponds and the total number of ponds.

We subsampled pond depths from survey flights on 30 June 2020 (pre-drainage) and 22 July 2020 (post-drainage). For each pond, we extracted the point furthest away from the pond edge as the pond center, the so-called pole of inaccessibility

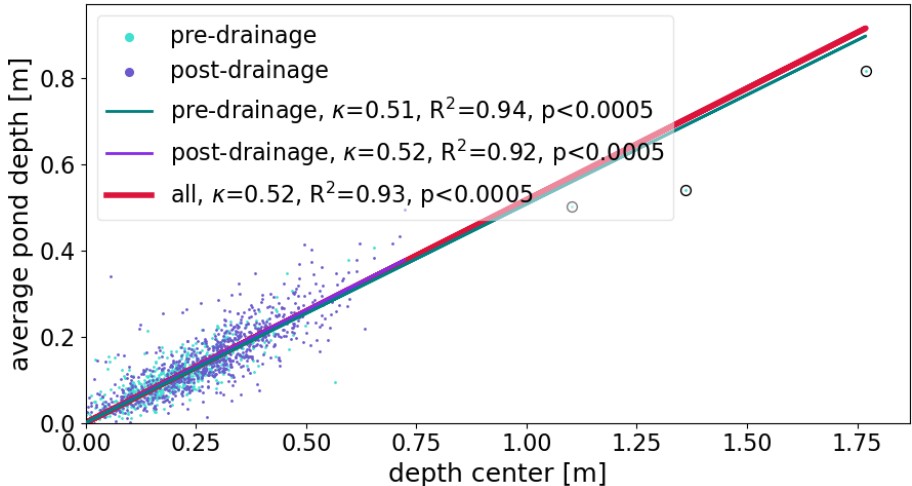

**Figure 17.** Linear dependency between single pond depth measurement at the pond center and mean pond depth reconstructed from aerial images of the MOSAiC leg 4 floe on 30 June 2020 (pre-drainage) and 22 July 2020 (post-drainage). Each dot represents a single pond. Particularly deep ponds were circled to make the dots more visible. All data was retrieved from aerial observations.

(PIA). We assumed that in situ depth measurements are typically or can be easily performed at this point, expecting the most representative depth in the center. All pond objects on the major floe classified with PASTA-ice were considered. Except to prevent errors caused by smoothing of the DEM in small ponds, ponds with a maximum distance between the center and pond edge of <1 m were neglected. In total, the evaluation was based on $1.6 \times 10^6$ pixels in 1621 aerially observed ponds. A selection between older (second and multi-year ice) and younger ice (first-year ice) was based on personal testimonies reporting 475 a younger ice area in the south of the floe (orientation of the floe in June/July 2020) and older ice in the center.

Comparing mean pond depth $d_{mean}$ derived from the entire pond bathymetries with single pond depth measurements in the center $d_{center}$ reveals that a single measurement in the center of the pond strongly overestimates the mean depth. On average, $d_{mean}$ is only 52 % of $d_{center}$ (Fig. 17). Hence, the mean pond depth and pond volume are much smaller than assumed based on single measurements from the pond's center. A descriptive form factor $\kappa$, which we define as

$$\kappa = \frac{d_{mean}}{d_{center}} = 0.52 \tag{7}$$

and retrieve from

$$\kappa = \frac{1}{n} \sum_{i=1}^{n} \left( \frac{d_{mean}}{d_{center}} \right)_i \tag{8}$$

including all $n = 1621$ ponds, shows strong statistical significance in the large data set. Residuals of the linear fit and thus deviations from this generalization are normally distributed (Fig. 18), independently if all ponds are considered (Fig. 18b), 485 or if subsets are taken for ponds in the pre- and post-drainage state (Fig. 18f) or on younger, less deformed (FYI) or older, strongly deformed (MYI) ice (Fig. 18d). Furthermore, residuals neither correlate with the pond area (Fig. 18a) nor the maximum disk area from which the single depth measurements are taken (Fig. 18c), nor the pound roundness (defined here as

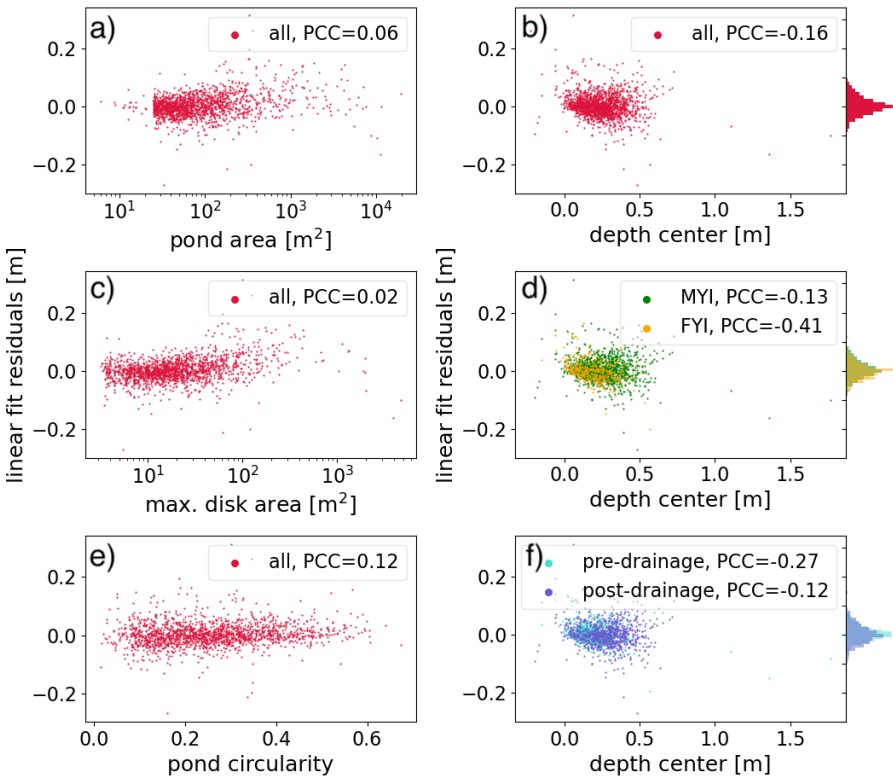

**Figure 18.** Residual analysis of the linear fit given in Fig. 17. Residuals of all ponds are shown with respect to their area (a), center depth (b), pole of inaccessibility (PIA) disk area (c), center depth and ice age (d), pond circularity (e), and center depth together with drainage stage (f). Histograms on the right side show the distribution function of the residuals (b) of all data, (d) separated by the ice age, and (f) separated by drainage stage. *pond circularity* is the ratio between the max. PIA disk area and pond area, indicating the shape complexity of the pond.

$\frac{\text{max. disk area}}{\text{pond area}}$) (Fig. 18e), nor the single depth measurement (Fig. 18b), nor the drainage state (Fig. 18f). Only ponds on young ice seem to develop a slight dependency of the form factor fit residuals from the pond depth with a Pearson correlation

coefficient of -0.41 (Fig. 18d). From this robust relationship between $d_{center}$ and $d_{mean}$ independent of the pond type, we conclude that the descriptive form factor $\kappa$=0.52 can be used to upscale single point measurements from the center of the pond (pole of inaccessibility) to mean pond depth.

The introduced form factor describes the average relationship between center depth and mean depth. For individual ponds, deviations may occur as indicated by the residuals of the fit. We explore the number of sampled ponds at which the form factor

becomes a valid estimator for the mean pond depth of multiple ponds. We calculated the form factor for 100 randomly chosen subsamples of different sizes $n$ from the 1621 ponds, similar to bootstrapping. The defined goal was that 99.3 % of the retrieved form factors within the 100 random samples are located within the range of $\pm$10 % of the form factor $\kappa$=0.52. Figure 19 shows the box-and-whisker plots for different pond sample sizes $n$, retrieved from the 100 random samples per size $n$. The upper and

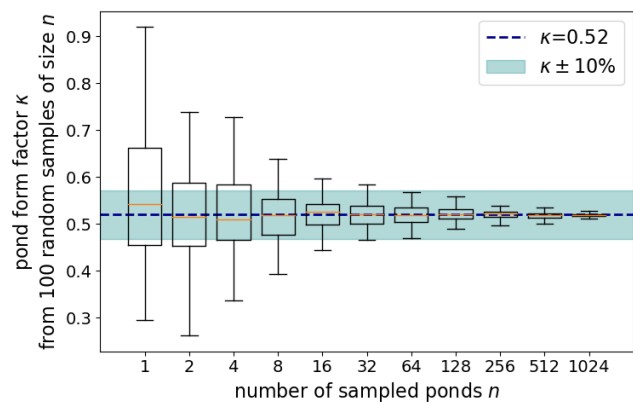

**Figure 19.** Box-and-whisker plots showing the mean (orange line), lower and upper quartile (box), and the minimum and maximum range (whiskers, 99.3 % of all data points inside this range) of bootstrap samples of the form factor $\kappa$ that we introduce to upscale single point measurements of pond depth to mean pond depth. Statistics are retrieved from 100 random samples of pond sample size $n$ from the entire set of 1621 ponds used in this study. The shaded area shows the targeted accuracy of the pond form factor as described in the text.

lower whiskers define the 99.3 % range. It can be seen that from 64 ponds on, the requirement is fulfilled. Thus, at least 64 ponds are needed to yield a valid approximation of the mean pond depths from single point measurements in the center of the ponds, applying the form factor $\kappa$.

## 6 Discussion

### 6.1 Photogrammetric reconstruction workflow for melt pond bathymetries

Photogrammetry can be used to reconstruct the sea ice surface topography from aerial images (Neckel et al., 2023; Divine et al., 2016). In the first part of this study, we investigated the ability also to reconstruct the full melt pond bathymetry. In particular, we have developed methods to consider water in the light paths used for reconstruction, in addition to air. We developed the methods using data from the PASCAL campaign. For this particular campaign and study site, more extensive processing was necessary, including corrections due to curved DEMs and the manual digitization of the pond edges. The evaluation of MOSAiC data showed that these were no longer required with better data quality through adapted flight patterns and better classifiable images. We thus present an algorithm that can be adjusted depending on the quality and method of the recorded data. In the following, we discuss the results obtained for the bathymetry reconstruction.

### 6.1.1 Impact of sky conditions, pond color, size, and shape

Within the presented dataset, no difference in the accuracy of the depth retrieval could be traced back to sky conditions or pond colors. However, we identified potentially larger uncertainties for very small and large ponds. Ponds smaller than 1 m in the horizontal expansion (equalling <10 times the ground sampling distance) were not properly reconstructable. We hypothesize

that the essential smoothing of the image depth maps flattens the topography in such small ponds. Further, on very large ponds, we assume that wind fetch causing ripples on the water surface could lead to disturbing reflections. In a weak form, this effect was observed on the above-mentioned *Mystery lake* (diameter approx. 80 m) on the MOSAiC floe, but it was not strong enough to degrade the retrieval quality.

In addition, unavoidable smoothing of the topography in the photogrammetric calculations may also lead to larger uncertainties for steep pond bottom slopes at the outer edges of ponds. Too few measurements were available to evaluate this quantitatively. However, we observed such effects by comparing the results visually to data presented in König et al. (2020). Since their method relies on spectral differences, it does not smooth the bathymetry toward the pond edges. Due to a lack of data, we could not test whether vertical pond walls that occur later in the season negatively impact the pond surface extraction
from the DEM. However, we assume that, in this case, the automatic smoothing reduces the error.

### 6.1.2    Accuracy of the in situ data and suggestions for future field campaigns

The accuracy of photogrammetrically derived pond depths was obtained compared to in situ measurements as ground truth. As discussed in König et al. (2020), those measurements also contain substantial sources for errors, namely metersticks that slip into cavities and the complex retracing of in situ measurement points in the airborne data. To compensate for the latter, we
averaged above a circular area with a radius of 0.3 m around the in situ points. We strongly recommend a sophisticated system for future campaigns to allow for a direct georeferencing link between in situ and airborne data. A pure GPS-based method proved to be insufficient for constantly drifting ice. Therefore, we recommend a system consisting of GPS base stations with regular freeboard measurements that are recognizable in images and record the geographic position in the Earth-fixed system. Such stations act as optical and geospatial GCPs, also improving the photogrammetric analysis through accurate horizontal and
vertical position reference. These reference stations must encompass the measurement area so that in situ measurements within the study area can be efficiently and accurately triangulated between these points. The best choice of tools, e.g., 360° cameras, theodolites, or low-range indoor positioning systems like Bluetooth beacons, needs to be tested. To reduce the error from meterstick point measurements caused by rippled or porous ice, we recommend the usage of a flat plate at the bottom of the depth gauge in future campaigns. This could, however, possibly lead to photogrammetric measurements slightly overestimating
the pond depth since the optical transition layer is probably located somewhat deeper than the upper haptic transition between pond water and ice. This requires further testing. It has also been shown on MOSAiC that measurements with echo sounders from remote-controlled boats can be used as a reference. Yet, even acoustic systems, especially in the high-frequency range of 500 kHz as implemented on the *Böötle* on MOSAiC, do not always detect the exact transition from water to ice as shown under laboratory test conditions (Werner, 2022). In general, we assume that in situ reference depths, both from manual and
acoustic measurements, are slightly overestimated by a few centimeters, if at all, and are, therefore, a suitable means of testing the algorithm for simultaneous flights. The acoustic measurements in this study, however, could only confirm the great depths in the observed ponds and derive trends, but no exact error could be derived due to the time lag of two days.

### 6.1.3 Required flight pattern

Our results suggest that pond depth studies require flight grids with great overlap of adjacent images. The more lateral offset is achieved in addition to forward overlap in the flight direction, the higher the accuracy of the pond depth measurements. However, simultaneously, we showed that small measuring angles, i.e., large aperture angles, should be avoided, as they lead to a greater underestimation of depths due to light refraction. This effect is most likely responsible for the observed discrepancies between the in situ (echosounder) and aerial pond depths in the deep parts of *Mystery lake* on MOSAiC, besides ongoing melting between the observing times. To achieve a high overlap and lateral offset at small measuring angles, aerial images must be captured with a high measurement frequency or low flight speed. To give an example for flight planning, the system we used on MOSAiC is limited to a measurement frequency of 0.25 Hz. To obtain a reasonable ground sampling distance of 5 cm, the camera resolution limits flight altitude to 100 m (300 ft). A circle with the maximum angle of incidence of the measurement of 40° that we defined to prevent errors in the photogrammetric reconstruction from light refraction then has a ground radius of 84 m. It is commonly recommended for photogrammetric measurements to use 80 % forward and 60 % lateral overlap in images for optimal reconstruction. Maximum flight speed would thus be limited to 4.2 m/s (8 kts) and the lateral offset of flight lines to 33.6 m. Fortunately, state-of-the-art aerial imaging systems can provide sufficiently higher temporal measurement frequencies to facilitate measurements.

In the photogrammetrically reconstructed topography, the DEM, large-scale gradients, also known as the doming effect, can typically occur. Since the same problem occurred with the PASCAL data, we have introduced tools that include a separate analysis of the ponds so that a derivation of the pond depth is still possible. The effect can be minimized by improving the calibration of the camera model through improved flight patterns, a slightly oblique camera perspective (Wackrow and Chandler, 2008), and, as we found when comparing the PASCAL and MOSAiC data, by not using an additional camera protection window in front of the lens.

## 6.2 MOSAiC melt pond bathymetry

The second part of the study applies the developed method to data from the MOSAiC campaign. Here, we discuss the major results from that unprecedented insight into temporal and spatial pond evolution and compare it to conventional measurement methods.

### 6.2.1 Pond bottom and level evolution

The large-scale and high-resolution observation of ponds on the MOSAiC floe points to several previously unconsidered melt-season processes. Firstly, large ponds on level ice in the deformed center of the MOSAiC floe were extraordinarily deep (>2 m) before the drainage event. They were thus as deep as the level ice on which they formed was thick (e.g., Itkin et al., 2023; Thielke et al., 2023; von Albedyll et al., 2022), while the pond bottoms themselves still consisted of 1 m and thicker ice, as anecdotal reports from the field showed. After drainage, pond bottoms, even of previously deep ponds, surfaced, although the pond level was close to sea level before. Both these observations are indications for a flexible ice cover (as shown in Fuchs,

2023c) and contradict the traditional assumption of a rigid ice cover on which ponds only reduce to their lowest points during the drainage event (e.g., Popović et al., 2020).

Secondly, it was mostly assumed or observed so far that pond surfaces are well above sea level before drainage (e.g., Polashenski et al., 2017; Perovich et al., 2021). This was the case only for a few small ponds in the deformed center of the floe. Only their bottom ice seemed impermeable and rigid enough on MOSAiC to resist the hydrostatic head that forms when ponds

are above sea level. Even before the drainage event, the largest pond with a water level >0.2 m higher than the sea level had an exposed water surface of 13.4 m meters in diameter (disk size around the pole of inaccessibility). 46 ponds on the MOSAiC floe were larger, but their pond level was closer to sea level on 30 June. This indicates that especially large ponds have little possibility of being far above sea level and that blockage processes reducing permeability (Polashenski et al., 2017) are not able to fully sustain the impermeability of the underlying ice. Instead, we speculate that a balance of water inflow and outflow

to ponds is established (lateral, vertical, and melting), allowing a pond level only minimally above sea level. Or, alternatively, that especially in large ponds the bottom ice can bend downwards under the load of accumulated meltwater and thus reduce the surface elevation, while the hydrostatic head, causing the bending, is still higher (as discussed in Fuchs, 2023c). However, more detailed investigations of the mechanical properties of the melting ice are required to break down the individual contributions of the processes to the observed results.

The presented observations of the temporal and spatial pond bathymetry evolution indicate that the pond bottoms can bend downwards under the weight of the accumulated meltwater and, after drainage, when the ice is very permeable, bend upwards due to the buoyancy of the ice. This flexibility must be considered when observing hydrostatic heads in the fields and parameterizing the pond development in models.

### 6.2.2 Representativeness of in situ studies

In situ transect studies are among the most common methods to collect representative in situ data during sea ice physical field campaigns. The entire sea ice column, including snow depth, pond depth, and ice thickness, can be probed on walked transect lines, providing comprehensive data on the properties of the atmosphere, snow, ice, and ocean and transition zones between them. However, already Perovich et al. (2003) noted that pond coverage along their transect line on the SHEBA campaign (Uttal et al., 2002) with a peak coverage of 40 % was not representative for the entire SHEBA site, for which they retrieved a

comparable smaller and more constant pond coverage of less than 24 % from aerial imaging (Perovich et al., 2002). Also, on MOSAiC, differences were found in the derived pond coverage between the walked transect line and high-resolution satellite imagery covering the entire floe (Webster et al., 2022). Webster and colleagues attributed a comparably smaller pond coverage along the transect line to its location close to the floe edge, where lateral runoff of meltwater is known to reduce pond coverage (e.g., Wright et al., 2020). This may also reduce pond coverage on smaller floes as observed by Divine et al. (2015) in the

marginal ice zone.

Collocated transect and airborne measurements are rare, as they are only possible with enormous effort on large-scale field campaigns or require high-resolution satellite imagery in clear sky conditions, which has become available only in recent years. For this reason, combinations of both, such as those listed previously, are rare, and we know of no study in which both methods

have been combined with a focus on comparing them instead of broadening the data source. Owing to the data available from Webster et al. (2022), we could, therefore, for the first time, include a systematic comparison of derived geometric pond properties between both measuring methods. The presented results have shown that in situ transect measurements show slightly greater average pond depths than airborne derived and were not entirely representative for the floe before drainage. Combining in situ transect data with high-resolution aerial imaging, therefore, yields a great potential for strengthening studies on floe-scale. The observed slightly greater depths can be expected from manual in situ measurements due to uneven pond bottoms mentioned above. However, we noticed that mean pond depths are apparently commonly derived by averaging depth profiles along single lines through ponds. We want to point out that using a single line to infer the average pond depth is mostly geometrically incorrect due to the mostly round shape of the ponds and possibly overestimates the mean pond depth. Shallower areas at the pond edges are underrepresented in such an extrapolation process.

Because of the possible overestimation of mean pond depth by in situ studies, we investigated the possibility of extrapolating mean pond depths of a large number of ponds from single in situ measurements at their center point. On average, the introduced form factor $\kappa$=0.52 indicated a strong correlation between pond depth at the center and mean pond depth. In the center, ponds were about twice as deep as on average, largely independent of other factors such as shape, depth, ice deformation, or stage of evolution. This result, collected from a sample size of 1621 ponds, could help improve mean pond depth estimations in the field where a complete manual assessment of the pond bathymetry is not feasible due to the limited workload, especially for a great number of individual ponds. Measuring only one depth per each of these ponds at the relatively easily findable spot pole of inaccessibility and estimating the mean pond depth using the pond form factor could massively increase the representativeness of the measurement for the mean pond depth.

### 6.2.3 Large scale pond observations including satellite data

Despite its proven broad applicability, the presented aerial image method for reconstructing pond geometry remains a spatially and temporally limited instrument that requires upscaling by satellite data for pan-Arctic observations. Our surface classifications show no noticeable difference compared to those with another algorithm (OSSP) applied on high-resolution satellite images of the same area (DigitalGlobe WorldView, Webster et al., 2022). Since we also see no further reason for a systematic difference between these high-resolution image input data, we assume that, apart from the different applicability (sky conditions, spatial and temporal coverage), both high-resolution surface class products compare well. Lower resolved optical satellite sensors require spectral unmixing techniques with temporally and spatially varying accuracy, which is discussed and evaluated in detail in other studies (e.g., Niehaus et al., 2023; Lee et al., 2024).

Regarding pond depths, novel approaches are emerging to upscale from a few manual point measurements on the ice to large-scale observations (Farrell et al., 2020; Herzfeld et al., 2023; Buckley et al., 2023; König et al., 2020). The ICESat-2-based observations offer enormous temporal and spatial coverage but are limited to larger ponds. Our high-resolution data can, therefore, make an essential contribution to evaluating and optimizing them. Similar to Buckley et al. (2023), we found that wide ponds defined here by their diameter at the pole of inaccessibility (which indicates the direction-independent extent of the pond shapes) contributed significantly to the overall melt pond volume on the floe. These ponds were exceptionally

deep in our data, which supports Buckley et al. (2023) statement that ICESat-2 data possibly overestimate the mean pond depth due to the reduced range in the observable melt pond size distribution. With the drainage event, the diameter and, with that, the direction-independent horizontal expansion of open pond water dropped significantly while the pond areas remained still high (caused by braided pond patterns) (section 5.1). This means that many drained ponds observed from the helicopter potentially fell below the minimum size thresholds of the ICESat-2 algorithms, which on the MOSAiC floe could have resulted in an underestimation of the bulk pond volume after drainage by up to 18.7 % to 68.4 % (DDA minimum width) or 92.3 % (UMD-MPA minimum width), assuming that no pond would have been observed from the satellite in a direction longer than that defined by the diameter. While the MOSAiC floe was certainly somewhat unique with its strongly deformed center part, this highlights how much small ponds (of which even the smallest are also cut off due to our 100-pixel threshold) contribute to the overall pond volume, especially after the drainage event when braided pond pattern reduce widespread, lake-like ponds. Incorporating data from the new aerial-based approach into satellite retrievals offers enormous potential for the upscaling of pond properties that should be fully exploited in the future.

## 7   Conclusions

We proved that pond bathymetry can be accurately derived from a photogrammetrical reconstruction of the ice surface. Aerial images from a monocular airborne camera in motion are sufficient, provided that strong overlap is given between single images. A simple multiplication of the derived water column depth with the refraction index of water (n=1.335) sufficiently corrects measured pond depths for light bending at the water–air interface. This factor naturally varies depending on the angle of incidence of the measurement. Still, here it has been shown that a constant factor is sufficient to be used subsequently to the complex bundle-block-adjustment in Agisoft Metashape, which favors nadir measurements. Incident angles at the surface were restricted to smaller than 40° to avoid horizontal alignment errors becoming greater than the ground sampling distance. Deviating from this limitation possibly contributed to a slight underestimation of large depths in the MOSAiC data.

With the newly developed method, we could reconstruct the evolution of pond coverage, depth, and volume on the MOSAiC leg 4 floe. The pond volume of the entire floe was more constant than found by Webster et al. (2022), who extrapolated it from a transect line around the floe. This difference was mainly caused by very deep ponds with large diameters in the floe center that doubled the mean area-specific pond volume of the floe before drainage when coverage was still slightly lower. Harmonization of the pond cover after drainage increased the representativeness of the transect area and re-distributed pond volume to ponds with smaller diameters (5 m to 20 m), which then contributed significantly (92.5 %) to the overall pond volume. Furthermore, we could detect clear indicators for a flexible ice cover below the ponds and derive a scaling factor for retrieving mean pond depths from single pond depth measurements in the field. We showed that pond depth and volume strongly depend on the pond diameter at the pole of inaccessibility of the pond shape. We therefore recommend this pond diameter as a valuable measure of pond characteristics.

The study showcases unexploited possibilities of aerial imaging of melt ponds. The developed methods and procedures allow pond bathymetry reconstruction solely from one optical camera deployed on a helicopter or airplane. The resulting availability

and exact alignment of optical and morphological data provide a unique database that requires only relatively inexpensive instrumentation and evaluation software to be compiled. Although we used the commercial software Agisoft Metashape for the most complex processing steps, we would like to point out that there are open-source projects like MicMac (Rupnik et al., 2017) available, which could further reduce the cost for such systems. Pre- and postprocessing is completely based on freely available Python and QGIS packages. The only change to previous campaigns, which eventually allows for the pond depth retrieval, is the appropriate design of the flight tracks. This approach is probably directly transferable to unmanned aerial vehicles (UAVs). Such a transformation from airplane-based observation to UAVs holds great potential for reduced emissions and an economically more friendly collection of sea-ice observations. The data acquired during MOSAiC shows that the method enables multidimensional studies tackling questions far beyond the current scope.

*Code and data availability.* Code and files are made freely available for further use:

- The surface classification tool PASTA-ice is accessible under https://github.com/nielsfuchs/pasta_ice and https://doi.org/10.5281/zenodo.7548469 (Fuchs, 2023c, a).

- Training data for PASTA-ice are available under DOI: https://doi.org/10.5281/zenodo.7513631 (Fuchs, 2023b).

- MOSAiC DEMs and Orthomosaics used for the pond depth retrieval are available under DOI: https://doi.org/10.1594/PANGAEA.949433 (Neckel et al., 2022, 2023).

- Example code for depth determination from a DEM raster and a classification Shapefile is available in the above-mentioned PASTA-ice repository under *helpful/PondDepth_retrieval/XX_process_ponds.py* (Fuchs, 2023a).

- The compiled pond bathymetry maps from MOSAiC are accessible on PANGAEA DOI: https://doi.org/10.1594/PANGAEA.964520 (Fuchs and Birnbaum, 2024).

*Video supplement.* For the EGU 2021 online meeting, we prepared an interactive online tour through the PASCAL study site, on which one can learn about the pond bathymetry determination (Fuchs et al., 2021). The tour is available under https://nielsfuchs.github.io/egu2021_pond_bathymetry_tour/.

## Appendix A: Calculations

The following geometric approaches were used to derive the horizontal and vertical mismatch caused by refraction in the photogrammetric depth calculations.

## A1 Horizontal mismatch

From Fig. 2, one can geometrically derive a set of equations describing the different sections of $X$:

$$X_{\beta_1} = \Delta X + X_{\alpha_1} \qquad\qquad\qquad\qquad = Z_{AP} \cdot \tan\beta_1 \tag{A1}$$

$$X_{\beta_2} = X_{\alpha_2} - \Delta X \qquad\qquad\qquad\qquad = Z_{AP} \cdot \tan\beta_2 \tag{A2}$$

$$X_{\alpha_1} = Z_{TP} \cdot \tan\alpha_1 \tag{A3}$$

$$X_{\alpha_2} = Z_{TP} \cdot \tan\alpha_2 \tag{A4}$$

Equating these with $Z_{AP}$ results in:

$$\Delta X = \frac{X_{\alpha_2} \cdot \frac{tan\beta_1}{tan\beta_2} - X_{\alpha_1}}{1 + \frac{tan\beta_1}{tan\beta_2}} \tag{A5}$$

By substituting $X_{\alpha_i}$ and applying Snell's Law to replace $\beta$ by the known angle of emergence $\alpha$, equation A5 becomes:

$$\frac{\Delta X}{Z_{TP}} = \frac{tan(\alpha_2) \cdot \frac{tan\left(arcsin\left(\frac{sin\alpha_1}{n_{water}}\right)\right)}{tan\left(arcsin\left(\frac{sin\alpha_2}{n_{water}}\right)\right)} - tan(\alpha_1)}{1 + \frac{tan\left(arcsin\left(\frac{sin\alpha_1}{n_{water}}\right)\right)}{tan\left(arcsin\left(\frac{sin\alpha_2}{n_{water}}\right)\right)}} \tag{A6}$$

For this, we use the simplified image of opposite measuring points and two different angles of emergence on the pond surface.

## A2 Vertical mismatch

The deviation in depth between measured and true depth is derived from geometric analysis of Fig. 2:

$$X_2 - X_1 = Z_{AP}tan\beta_2 + Z_{AP}tan\beta_1 \tag{A7}$$

$$X_2 - X_1 = Z_{TP}tan\alpha_2 + Z_{TP}tan\alpha_1 \tag{A8}$$

Equating both and including Snell's Law leads to:

$$\gamma = \frac{Z_{AP}}{Z_{TP}} = \frac{tan\alpha_1 + tan\alpha_2}{tan\beta_1 + tan\beta_2} \tag{A9}$$

$$\gamma = \frac{tan\alpha_1 + tan\alpha_2}{tan\left(arcsin\left(\frac{sin\alpha_1}{n_{water}}\right)\right) + tan\left(arcsin\left(\frac{sin\alpha_2}{n_{water}}\right)\right)} \tag{A10}$$

We apply a limit value analysis for small incident angles. To do so, we first equate both measuring angles $\alpha_1 = \alpha_2$. This results in a simplified form of equation A10:

$$\gamma = \frac{tan\alpha}{tan\left(arcsin\left(\frac{sin\alpha}{n_{water}}\right)\right)} \tag{A11}$$

In this form, the equation is undefined at zero since $tan(0) = 0$ and $arcsin(0) = 0$. However, with a couple of rearranging tricks, one gets an evaluable formula form.

Inserting:

$$tan(arcsin(x)) = \frac{x}{\sqrt{1-x^2}} \tag{A12}$$

into equation A11 leads to:

$$\gamma = \frac{tan(x)}{\left( \dfrac{\frac{sin(\alpha)}{n_{water}}}{\sqrt{1-\left(\frac{sin(\alpha)}{n}\right)^2}} \right)} \tag{A13}$$

by rearranging and the definition of the tangent we obtain:

$$\gamma(\alpha) = cos(\alpha) \cdot \sqrt{n_{water}^2 - sin^2(\alpha)} \tag{A14}$$

## Appendix B: MOSAiC pond properties resolved by pond area

*Author contributions.* N.F. developed the methods and the concept of the study, analysed the data, and wrote the paper, with contributions from all authors. G.B. conceived the aerial imaging platform. N.F., G.B., F.L., N.O., L.v.A. collected measurement data.

*Competing interests.* At least one of the (co-)authors is a member of the editorial board of The Cryosphere.

*Acknowledgements.* Data used in this manuscript was collected as part of RV *Polarstern* (Knust, 2017) campaigns PS106-1 PASCAL (Project ID AWI_PS106_00), project TEMPO (Oppelt et al., 2017), and the Multidisciplinary drifting Observatory for the Study of the Arctic Climate (MOSAiC) with the tag MOSAiC20192020 (Project ID AWI_PS122_00). We cordially acknowledge the helicopter and ship crew of RV *Polarstern* and Marcel König for their support in collecting the measurement data.

Major parts of the study were developed based on the published PhD thesis *A multidimensional analysis of sea ice melt pond properties from aerial images*, by Niels Fuchs, 2023, University of Bremen, Germany (Fuchs, 2023c).

N.F. acknowledges the support of the AWI and funding by the BMBF project NiceLABpro (03F0867A). L.v.A. acknowledges the support of the AWI through its project AWI_ICE and the Deutsche Forschungsgemeinschaft (DFG) through the International Research Training Group IRTG 1904 ArcTrain (grant 221211316) and her ESA CCI research fellowship ACCURATE. N.O. and F.L. were supported financially by the German Federal Ministry for Economic Affairs and Climate Action (BMWK, funding code: 50EE1917A) and the Prof. Dr. Werner Petersen Stiftung, Kiel, Germany.

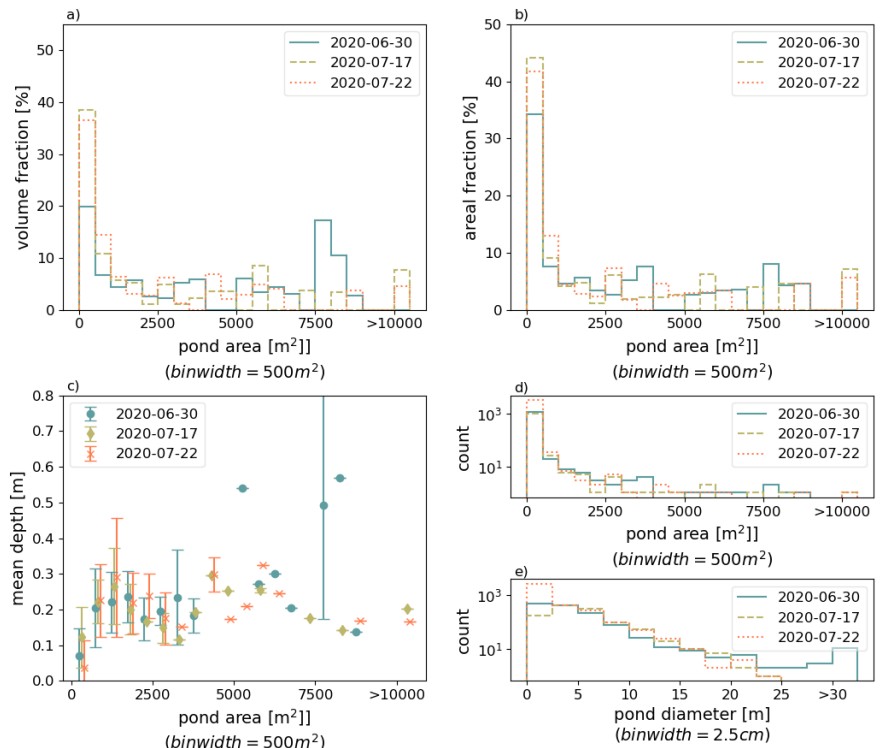

**Figure B1.** Distribution functions of pond volume (a), pond areal fraction (b), mean pond depth (c) and number of ponds (d) in 21 pond area bins from 0 m$^2$ to 10000 m$^2$ and larger than 10000 m$^2$ on the MOSAiC floe. (e) shows the number of ponds resolved by pond diameter as used in Figure 15. Shown are data retrieved from photogrammetric reconstructions from aerial images collected on survey flights on 2020-06-30 (pre-drainage) and 2020-07-17, 2020-07-22 (post-drainage).

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
