# Peer review of "Sea ice melt pond bathymetry reconstructed from aerial photographs using photogrammetry: A new method applied to MOSAiC data."

_EGUsphere, 2023_

## Referee Comment (RC1)

Review of the manuscript "Sea ice melt pond bathymetry reconstructed from aerial

photographs using photogrammetry…" by Fuchs et al.

In this study the authors apply a photogrammetric technique for reconstructing sea ice surface topography with the focus on melt pond bathymetry. For method elaboration and testing results of two field summer campaigns in the Arctic, including Mosaic are used. The major novelty of the proposed approach is a correction factor to account for refraction at air-pond surface interface incorporated in the data processing and analysis workflow. Authors further analyse the derived DEM of melt ponds providing basic statistics on their geometry, and using bootstrapping assess a representativeness of pond depths from in situ measurements for pond water/volume estimates. Proposed correction factor can then be used for more accurate calculations of melt pond volumes from in situ measurements.

I am generally very positive about the presented work. Since I did myself a similar study some years ago, I am very much familiar with the scales of technical work/various challenges/computations required for this kind of study. There are only a few minor remarks/suggestions I would like the authors to address, both technical and regarding the analysis of the data.

Line 141: Should it be just 2x2 km^2 ?

Line 146: "Photogrammetrically reconstructed DEMs from 30 June 2020 and 22 July 2020 were leveled to zero water level using a flat plane fitted trough all lateral snow/ice‑open water boundaries positions in the DEM within the cropped…"

I wonder if these Z-control points were selected manually or identified automatically from "melt ponds" objects that fell into the edge?

I have a number of technical comments to Section 3, more to satisfy my curiosity need to say.

What lense correction model the authors have used? The "standard" one available in Agisoft or you ran a target based calibration specifically on the camera and the lens used in the setup? Also, how did you disable the autofocus (just curious, since any movement of lenses changes the optical parameters of the camera system, this must be made rigid in one or another way).

I wonder also if the authors worked with the raw image format or compressed jpgs? One of my challenges (some 10 years ago) was linked with a computational intensity of the entire process due to dealing with tiffs in the original resolution. How computer intensive, in general, the process was in your case. Did Agisoft manage to "digest" the entire "Fortress" in one go or you had to break the scene in pieces? How many images in total where involved into a bundle when building the "Fortress" DEM?

I also noticed that the authors did not apply any ice drift correction to camera positions prior to triangulation. From my experience the drift of sea ice causes the emergence of scene-scale gradients in the reconstructed DEM, but I assume, "forcing" the edges of the derived Fortress DEM through the plane could have helped to resolve the problem.

Line 263: Can you please discuss, how would it work with well elaborated melt ponds later in the season, when nearly vertical walls of the ponds with some 10-15 cm freeboard could emerge by melting?

Line 294: Did you actually run the classifier first with all 9 subclasses and then combined, or you used 3 classes only directly?

Line 336: Are there any more details already published on this vast melt pond? Appears to be a rather unusual object, for the pond to be that deep.

Line 379: I do suggest referring to the Discussion section here where this phenomenon is discussed and the likely explanation proposed, as such a drastic change in elevation (over 1.5m!) immediately grabs attention.

Fig.14 caption: Please add grade shade scale bar for surface elevation (above 0). BTW, did you try to compare DEM from ALS and photogrammetric DEM from this study?

Line 388: What is the contribution of these two largest ponds into the total meltwater/pondwater budget? In general, one can consider making a pdf of pond sizes/pond water volume in order to see which ponds contribute most to the overall pond water budget.

Line 446: "young ice" or "FYI?"

Line 490: Good also to have the elevation (freeboard) measured at these GCPs too , close to the timing of overflight. My experience show that even without accurate XY GCPs, Z-control points already improve the accuracy greatly at they "force" the DEM into their proper position eliminating the elevation gradient.

Line 557: The effect of reducing pond coverage was also observed in Divine et al., 2015 (https://doi.org/10.5194/tc-9-255-2015) when melt pond fraction declined towards the edge of the MIZ due to decreasing floe sizes and hence stronger lateral drainage.

---

## Referee Comment (RC2)

General comments:
This study focusses on an algorithm for retrieval of melt pond bathymetry from photogrammetry, and comparison with in-situ validation data. This retrieval is independent of sky conditions and pond colour, and provides correction for refraction at the pond surface interface. Descriptions of ice conditions for context are notably well detailed. Retrieved pond bathymetries are analysed and discussed. I particularly enjoyed reading the discussion section, which is very clear and well-constructed. Code and data availability sections link to a notably very well documented code with a logical structure. This is a very good manuscript presenting some innovative and excellent work, and should be published subject to minor modifications to enhance clarity.

Specific comments:
Line 26:
Suggest replace 'simplified' with 'simplistically'.
Line 30:
Seems a shame to only mention the means and not to include the standard deviations reported in Morassutti and Ledrew (1996), Table 6. I think these would add to the already compelling case of the importance of your research, but I leave this to your judgement.
Line 50:
The standard convention is 'ICESat-II', not 'IceSAT-2'
Line 66:
Make clear the type of survey you're referring to.
Figure 1:
Appreciate the lat/lons for the study area are provided in text but could you please also include them in the figure caption.
Line 295:
Good interpretation of accuracy values.
Figure 2:
"All variables are explained in the text." This is not sufficient, Please add to the caption variable definitions.
Figure 12:
To clarify, is the penultimate class (royal blue) indeed >35cm or 35<x<50? I wonder if a colour bar may be more appropriate here.

---

## Author Comment (AC1)

**Author response to the review of the manuscript "Sea ice melt pond bathymetry reconstructed from aerial photographs using photogrammetry…" by Dmitry Divine**

Black: Comments from the reviewer
Red: Responses from the authors

We were thrilled to receive such a positive assessment from the reviewer, who has proven experience in photogrammetric sea ice studies. The questions demonstrate a thorough understanding of the applied methods. We are pleased to be able to provide answers and have carefully considered all comments in the revised version, including minor adjustments that include the answers to technical questions to make them accessible to future readers of the study.

Line 141: Should it be just 2x2 km^2 ?

Many thanks for making us aware of this error. We changed it to 2 km x 2 km (here and later)

Line 146: "Photogrammetrically reconstructed DEMs from 30 June 2020 and 22 July 2020 were leveled to zero water level using a flat plane fitted trough all lateral snow/ice–open water boundaries positions in the DEM within the cropped…
"I wonder if these Z-control points were selected manually or identified automatically from "melt ponds" objects that fell into the edge?

We added: "These reference points were automatically extracted from the raster data DEM at the positions of touching surface class vector polygons." to make this clearer and adjusted the paragraph to embed the modification.

I have a number of technical comments to Section 3, more to satisfy my curiosity need to say.
What lense correction model the authors have used? The "standard" one available in Agisoft or you ran a target based calibration specifically on the camera and the lens used in the setup?

We tried target-based calibration; however, we found during test flights over earth-fixed ground (Emden, Germany) with constant aperture that surface reconstruction still worked better with initialized but free parameter adjustment in the Agisoft camera correction model. Apparently, temperature, small changes in focus (see below), and other points impacted the calibration.

Also, how did you disable the autofocus (just curious, since any movement of lenses changes the optical parameters of the camera system, this must be made rigid in one or another way).

We adjusted the focus before the flight in the horizontal view to a point approximately at the distance of the flight height. We then completely switched off the autofocus, both electrically and mechanically, and secured the adjustment wheel on the lens with adhesive tape.

We added, "All aerial images of a survey flight were taken with constant exposure settings and a mechanically and electrically fixed autofocus set to the flight altitude during flight preparation." And "We use the commercial photogrammetry suite Agisoft Metashape to calibrate the camera optics and solve the complex aerial triangulation equations to calculate orthomosaics and DEM as georeferenced raster data." to the method section.

I wonder also if the authors worked with the raw image format or compressed jpgs? One of my challenges (some 10 years ago) was linked with a computational intensity of the entire process due to dealing with tiffs in the original resolution. How computer intensive, in general, the process was in your case. Did Agisoft manage to "digest" the entire "Fortress" in one go or you had to break the scene in pieces? How many images in total where involved into a bundle when building the "Fortress" DEM?

The full pre-processing of the used MOSAiC orthomosaics is described in Neckel et al. (2023) (https://doi.org/10.1038/s41597-023-02318-5). We used full resolution images converted to JPG before processing and could reconstruct even much larger areas than the Fortress in one go (from >2000 images). The efficiency of photogrammetry suites has massively increased over the last few years.

I also noticed that the authors did not apply any ice drift correction to camera positions prior to triangulation. From my experience the drift of sea ice causes the emergence of scene-scale gradients in the reconstructed DEM, but I assume, "forcing" the edges of the derived Fortress DEM through the plane could have helped to resolve the problem.

This is a very good point and a well-known problem of photogrammetric DEMs, whereby such large-scale gradients and bending can occur not only with drifting sea ice but also very distinctly over land. The so-called "Doming effect". Its impact on our data is mentioned in section 3.5 of the manuscript. On PASCAL, we analyzed each pond separately to reduce the impact of large-scale gradients. This point was important for us to mention, as it also shows future users how to deal with it. Since your question showed us once again the importance of the topic, we have touched on it again in the discussion 6.1.3:
"
In the photogrammetrically reconstructed topography, the DEM, large-scale gradients, also known as the doming effect, can typically occur. Since the same problem occurred with the PASCAL data, we have introduced tools that include a separate analysis of the ponds so that a derivation of the pond depth is still possible. The effect can be minimized by improving the calibration of the camera model through improved flight patterns, a slightly oblique camera perspective (Wackrow and Chandler, 2008), and, as we found when comparing the PASCAL and MOSAiC data, by not using an additional camera protection window in front of the lens.
"

Learning from PASCAL, gradients were already much smaller on MOSAiC. However, the remaining very large-scale gradients in the MOSAiC DEMs were corrected in comparison to the available airborne laser scanner (ALS) data (Neckel et al., 2023). On days with high ice drift speeds, camera positions were additionally corrected, but mainly to improve and speed up the alignment.

Line 263: Can you please discuss, how would it work with well elaborated melt ponds later in the season, when nearly vertical walls of the ponds with some 10-15 cm freeboard could emerge by melting?

Thanks for mentioning this. We expect that the automatic smoothing of the reconstructed topography minimizes the effect. We added that point to the method description:
"Then, we extracted the relative height of the pond surface $h_{surf}$ at the pond margin from the DEM, which marks the transition from ice to pond in the smoothed topography. Due to the smoothing, we expect the method to be valid also for ponds with almost vertical walls later in the season (Fetterer and Untersteiner, 1998), which were not part of the evaluation set, however." However, as we are unable to test this, we decided to include it in the discussion as a possible uncertainty: "Due to a lack of data, we were also unable to test whether vertical pond walls that occur later in the season negatively impact the pond surface extraction from the DEM. However, we assume that in this case, smoothing reduces the error."

Line 294: Did you actually run the classifier first with all 9 subclasses and then combined, or you used 3 classes only directly?

We ran it with all 9 subclasses as tests in Fuchs, 2023, showed, that for the main classes, there is no gain or loss in accuracy if classified directly or from subclasses. We slightly adjusted the passage in the text to make this clearer: "Pixels in orthomosaics are classified into nine different sea ice surface sub-classes that belong to three main classes (Table 3): snow/ice, open water and ponds (including submerged ice). Adjacent sub-class pixels of similar main classes are subsequently combined into main class vector objects if these consist of, at minimum, 100 pixels (the threshold was chosen similar to Huang et al., 2016)."

Line 336: Are there any more details already published on this vast melt pond? Appears to be a rather unusual object, for the pond to be that deep.

It is indeed a vast pond. Therefore, it was especially interesting as a test site for the algorithm. This manuscript is the first peer-reviewed study to investigate its depth (besides the PhD thesis of Fuchs, 2023). It is mentioned in Calmer et al. 2023, but only in the context of its Albedo, and shown in Figure 4 of Webster et al. 2022, but in the context of pond coverage evolution. We already started to investigate its depth evolution (e.g, IGS Conference 2023 contribution, discussion section in this paper) and decided that the full analysis of the underlying physical processes is worth a separate publication.

Line 379: I do suggest referring to the Discussion section here where this phenomenon is discussed and the likely explanation proposed, as such a drastic change in elevation (over 1.5m!) immediately grabs attention.

Great idea, we added: "We discuss this significant loss in depth of the largest ponds (incl. Mystery lake) in the discussion section."

Fig.14 caption: Please add grade shade scale bar for surface elevation (above 0). BTW, did you try to compare DEM from ALS and photogrammetric DEM from this study?

We clarified in the caption that grey areas show the orthomosaics; pond depth data are only overlaid. Based on the other reviewers' comments on figure captions, we revised all figure captions.

Photogrammetrically derived DEMs were not compared to ALS data here, since Neckel et al, 2023 used the ALS data to slightly adjust the photogrammetric DEMs of MOSAiC we used. A comparison can be found in the mentioned study. We have included this additional adjustment due to the data availability from MOSAiC. However, the data from PASCAL show that the pond depth determination works also purely with photogrammetry.

Line 388: What is the contribution of these two largest ponds into the total meltwater/pondwater budget? In general, one can consider making a pdf of pond sizes/pond water volume in order to see which ponds contribute most to the overall pond water budget.

Many thanks for this very nice idea. In response to your suggestion and another reviewer's comment, we decided to incorporate pond volume and sizes on MOSAiC even more into the manuscript. We added a new figure to the result section, adjusted the result description and with that, we have added a new aspect to the discussion on satellite upscaling methods, as these are closely related to the size distributions. We are convinced that this additional information is worth the slight extension of the manuscript by one paragraph and does not change any methodological aspects.

Line 446: "young ice" or "FYI?"

Many thanks for noting that. We kept the definition unspecific, as the separation is based only on personal testimonies. However, it was currently not even specific in being unspecific. We corrected that.

Line 490: Good also to have the elevation (freeboard) measured at these GCPs too , close to the timing of overflight. My experience show that even without accurate XY GCPs, Z-control points already improve the accuracy greatly at they "force" the DEM into their proper position eliminating the elevation gradient.

That's a very good point. We added that to the paragraph: "Therefore, we recommend a system consisting of GPS base stations with regular freeboard measurements that are recognizable in images and record the geographic position in

the earth system. Such stations act as optical and geospatial GCPs, also improving the photogrammetric analysis through accurate horizontal and vertical position reference."

Line 557: The effect of reducing pond coverage was also observed in Divine et al., 2015
(https://doi.org/10.5194/tc-9-255-2015) when melt pond fraction declined towards the edge of the MIZ due to decreasing floe sizes and hence stronger lateral drainage.

Many thanks for that great hint. We added a sentence to the discussion: "This may also reduce pond coverage on smaller floes as observed by Divine et al. (2015) in the marginal ice zone."

---

## Author Comment (AC2)

**Author response to the review of the manuscript "Sea ice melt pond bathymetry reconstructed from aerial photographs using photogrammetry…" by an Anonymous Referee**

Black: Comments from the reviewer
Red: Responses from the authors

We greatly appreciate the positive feedback and are very grateful for the modifications raised by the reviewer, which we have fully considered to improve the quality of the manuscript.

Specific comments:

Line 26:
Suggest replace 'simplified' with 'simplistically'.

Thank you for pointing out this incorrect use of words. We changed it accordingly.

Line 30:
Seems a shame to only mention the means and not to include the standard deviations reported in Morassutti and Ledrew (1996), Table 6. I think these would add to the already compelling case of the importance of your research, but I leave this to your judgement.

Many thanks for this great suggestion. We added the standard deviations and revised the entire paragraph according to the other reviewers comment.

Line 50:
The standard convention is 'ICESat-II', not 'IceSAT-2'

Thanks for noticing. We changed it to "ICESat-2", as mentioned by the other reviewer and used in the papers cited in this regard.

Line 66:
Make clear the type of survey you're referring to.

Many thanks for spotting this gap. We added: "Only one experimental study on the photogrammetric derivation of pond depth from aerial images was carried out above sea ice before by […]"

Figure 1:
Appreciate the lat/lons for the study area are provided in text but could you please also include them in the figure caption.

Many thanks for mentioning that. According to the other reviewers' comments about frequently too short figure captions, we have revised all of them, and in the course of this, we have also taken your suggestion into account.

Line 295:
Good interpretation of accuracy values.

We appreciate the feedback. Such high values of accuracy can be truly misleading!

Figure 2:
"All variables are explained in the text." This is not sufficient, Please add to the caption variable definitions.

Because of your comments on a couple of captions and the other reviewer's general comments on the caption, we revised all of them and added variable definitions to the captions to make the figures stand alone. Many thanks for making us aware on that.

Figure 12:
To clarify, is the penultimate class (royal blue) indeed >35cm or 35<x<50? I wonder if a colour bar may be more appropriate here.

Many thanks for noticing this flaw in the legend (the royal blue class was indeed 35<x<50). We gladly followed the suggestion and changed the style to a colorbar.

---

## Author Comment (AC3)

Author response to the review of the manuscript "Sea ice melt pond bathymetry reconstructed from aerial photographs using photogrammetry..." by Ellen Buckley

Black: Comments from the reviewer

Red: Responses from the authors

We greatly appreciate the positive feedback and rating of the manuscript. The reviewer, who has great experience in the field, has gratefully read through the manuscript carefully and brought up points that we have all considered in the revision to significantly improve its quality. We are very grateful for this in-depth review and have thoroughly proofread the manuscript.

General:

The Figure captions are pretty short, and sometimes say "described in the text" I recommend adding text to the figure captions so that they can be stand-alone figures. I imagine this will be a useful reference text for future studies and the description of diagrams in figure captions will make the methodology clearer.

We are very grateful for this positive feedback and have checked all captions and adjusted them where necessary.

Specific:

Line 26: awkward phrasing- maybe you mean "rather simply"

According to the other reviewer's suggestion, we changed it to "simplistically"

Comments on Line 29 And Line 42: I disagree that 'most melt pond depth obs. for models were published in Morassutti and Ledrew'. Melt pond depth measurements from SHEBA (Perovich et al., 2003, Figure 11) were used in CCSM4 parameterizations (Holland et al., 2012). You actually mention this in line 40. These thoughts could be combined.
I don't know if this is based on Luthje 2006. See Holland et al 2012- directly references Perovich et al. 2003 and the SHEBA measurements. Maybe this is true for the Pedersen scheme but certainly no all the links between pond fraction and depth

Many thanks for bringing this unspecific formulation to our attention. We rearranged the paragraphs to clarify the applications of the datasets and models.

Line 34. I would say "Here we define pond bathymetry as…" because some studies will refer to bathymetry as a two-dimensional sample instead of the whole bathymetric floor.

Thanks for this good specification. We changed it to: "However, the actual pond bathymetry, which we define here as the pond depth profile in all directions, and which therefore also yields the actual average pond depth, remains largely undiscussed in the literature."

Line 50: ICESat-2 (correct capitalization)

We changed it accordingly.

Line 50: Please also include the larger study by Buckley et al., 2023 (Follow on to Farrell et al., 2020) which involves two algorithms (also include Herzfeld et al., 2023 that described the DDA algorithm) to automatically retrieve melt pond depth applied to thousands of ponds in the 2020 melt season. Still not a comprehensive database but showcases the ability to retrieve pond depths at large scales. I suggest you also include this in the discussion section about pond depth and coverage (fraction) evolution.

Many thanks for pointing out these further developments of the ICESat-2 algorithms. We decided to incorporate them stronger into the revised manuscript by mentioning them in the introduction, adding a full paragraph on satellite upscaling to the discussion, and adding a new figure to the result section that displays volume and depth distributions (also in response to your comment on line 398 and the comment of another reviewer). We are convinced that this additional information is worth the slight extension of the manuscript by one paragraph and does not change any methodological aspects.

Line 51. Consider "Orbital path" instead of "flight track lines"

We changed it to "along the ground tracks of the satellite beams." as we are referring specifically to the track on the ice.

Line 54. Chiroptera flew over sea ice for the ICESat-2 summer validation campaign in 2022 and that is a ALB system. Not sure of any publications that include that information right now though.

Many thanks for spotting this deprecated information. We couldn't find any publications either, just conference abstracts. So, we changed it to: "To our knowledge, such an ALB system over sea ice was only deployed for the first time in 2022 as part of the ICESat-2 validation. "

Figure 1 caption. "Know" to "known"

Changed

Figure 1. can you make sure the arrows are contained within the image – it is hard to tell what they are.

White arrows are connected to the inlets. We reformulated the caption to make this point clearer.

Line 87. I'm confused about the use of "we." You are not the Macke and Flores authors- do you mean they did that? Or the authors on this paper also happened to be on the Polarstern cruise. The second half of this paragraph is in third person. Consider clarifying or putting the whole methods section in third person.

Thank you very much for bringing this to our attention. We have reworded it to clarify our active participation.

Line 101. Replace "most probably" with "most likely" or "likely"

Changed to "most likely"

Line 104: why can't these ponds be designated strictly as melt ponds?

The motivation behind the sentence was the likely first formation by flooding. However, we noted that it can be easily removed to avoid confusion.

Line 111: Reference for cloudy days being more common in Arctic summer?

„Weather conditions on 14 June (Fig. 1b) were exactly the opposite. The entire sky was covered by a stratiform cloud cover, as is usual in central Arctic summers (Cotton et al., 2011) when the average cloud coverage reaches its maximum of about 70% (e.g., Wang and Key, 2005)."

Line 123: change "reached" to "ranged" and "pond depth" to "pond depth measurements"

Check

Line 126: Can you quantify pond coverage increase?

Due to the constrained size of the measurement site in the ridge area, we describe this change only qualitatively, as a quantitative analysis would require an arbitrary definition of the study site size.

Line 134: either leave out "pandemic related" or include COVID-19 (hopefully people in 100 years will be reading this paper and may not know what this is referring to)

Haha, fingers crossed. We changed it to "because of an inevitable crew exchange on Svalbard".

Line 135: I think you said this somewhere else but can you remind us the age of the new ice floe?

We added that information.

Line 141: 2 km x 2 km you mean?

Many thanks for noting this mistake. We changed it and later occurrences

Line 146: how much error does this introduce?

Since we're using a flat plane to remove the vertical offset of the DEM from sea level, the standard deviation remains unaffected. However, the offset reduced from 0.49m to 0.05m on 2020-06-30 and from 1.11m to 0.04m on 2020-07-17. Due to the lack of further reliable data (such as a large number of ground control points), we cannot quantify the error further.

Line 209: can you make Snell's law a proper numbered equation in the text, or refer to Eqn. 1 here.

We added a reference to the previous equation.

Line 255: (e.g., Hutter et al., 2023) – im sure there are others so add the e.g.

Perfectly correct, we added „e.g.,"

Line 267: (Jordahl et al., 2020; Perry, 2015; Gillies, 2013). Either but these in chronological order or if these refer to the python libraries in order add ",respectively"

Thanks for the suggestion. We added "respectively"

Line 276: is the algo description in Fuchs 2023a or Fuchs 2023c- both are included here- what is the difference

Thanks for pointing towards this unclear formulation. We added a note, that the source code is available in Fuchs 2023a, while methods are described and evaluated in 2023c.

Line 279: perhaps here can you list the main classes and not just refer to Table 3

Thanks for the suggestion. We added that.

Line 301: if you include the QGIS version here, you should include it everywhere

Thanks for spotting this inconsistency. We removed the version information here as it contains no essential information.

Figure 11: Consider a gray or dark background so the light points stand out.

Many thanks for noting this possible ambiguity in the interpretation. We have adjusted both the color scale and the caption accordingly.

Line 370: Again the pandemic comment

Removed as previously.

Line 376: Sentence doesn't make sense – especially the ending "with partially more than 2 m)

We changed it to "On 30~June, several very large, exceptionally deep ponds (>2~m) had formed on the MOSAiC floe, along with many smaller ponds"

Line 381: relatively high underestimation is a confusing statement

Changed to "strong underestimation"

Line 385: does vertically elevated mean above sea level?

Changed to above sea level

Line 398: this is confusing. Pond volume across the floe was constant (in space or time?) And what do you mean "has not changed much either"… has not changed since when? And then the next clause you say the ponds do deepen?

Many thanks for making us aware of this unclear formulation. To make this point clearer, and also in response to the other reviewer's question about distribution functions, we have decided to add a figure that shows these relationships more clearly and provides an overview of the pond volume, depth, and area evolution on the MOSAiC floe. Consequently, we modified the section in the results and referred to it again in the discussion, where we also linked it to the newly added discussion on satellite upscaling (as previously mentioned in response to the earlier comment on the Buckley et al., 2023 paper).

Line 404: what do you mean both Webster et al., 2022? If you are talking about the ossp classification in webster, cite webster in the first set of brackets with Wright and Polashenski

Thanks for noting. We made this more precise: "In the following, we compare the aerial derived data to available results from high-resolution satellite observations (Webster et al., 2022, using the Wright and Polashenski (2018) classification algorithm OSSP) and in situ transect lines (Webster et al., 2022) to assess the accuracy of our results and the representativeness of observed areas."

Line 410: can you describe what you see in Fig 15a and quantify how well they match.

We added a more specific description of the differences between the satellite-, aerial-, and in situ-derived pond coverage evolution.

Figure 15: can you explain the recalculation, has this been fixed in the Webster manuscript? If so can you reference the correction?

The calculation used a wrong factor. We confirmed again that the journal is working on the correction. If it is completed by the time of final publication, we will include it here.

Line 476: all other instances of Mystery Lake do not have lake capitalized. Be consistent.

Changed to Mystery lake

Line 479: too few

Changed

Line 525: sentence starting with "most interestingly…" does not make sense.

We agree that this was unnecessary wording. We deleted it.

Line 545: "the here" doesn't make sense

We removed "here"